# TROPOMI/S5P formaldehyde validation using an extensive network of ground-based FTIR stations

Corinne Vigouroux[1], Bavo Langerock[1], Carlos Augusto Bauer Aquino[2], Thomas Blumenstock[3], Zhibin Cheng[4], Martine De Mazière[1], Isabelle De Smedt[1], Michel Grutter[5], James W. Hannigan[6], Nicholas Jones[7], Rigel Kivi[8], Diego Loyola[4], Erik Lutsch[9], Emmanuel Mahieu[10], Maria Makarova[11], Jean-Marc Metzger[12], Isamu Morino[13], Isao Murata[14], Tomoo Nagahama[15], Justus Notholt[16], Ivan Ortega[6], Mathias Palm[16], Gaia Pinardi[1], Amelie Röhling[3], Dan Smale[17], Wolfgang Stremme[5], Kim Strong[9], Ralf Sussmann[18], Yao Té[19], Michel van Roozendael[1], Pucai Wang[20], and Holger Winkler[16]

[1]Royal Belgian Institute for Space Aeronomy (BIRA-IASB), Brussels, Belgium
[2]Instituto Federal de Educaçao, Ciência e Tecnologia de Rondônia (IFRO), Porto Velho, Brazil
[3]Karlsruhe Institute of Technology (KIT), Institute for Meteorology and Climate Research (IMK-ASF), Karlsruhe, Germany
[4]German Aerospace Centre (DLR), Remote Sensing Technology Institute, Oberpfaffenhofen, 82234 Weßling, Germany
[5]Centro de Ciencias de la Atmósfera, Universidad Nacional Autónoma de México (UNAM), Mexico City, México
[6]Atmospheric Chemistry, Observations & Modeling, National Center for Atmospheric Research (NCAR), Boulder, CO, USA
[7]Centre for Atmospheric Chemistry, University of Wollongong, Wollongong, Australia
[8]Finnish Meteorological Institute (FMI), Sodankylä, Finland
[9]Department of Physics, University of Toronto, Toronto, Canada
[10]Institut d'Astrophysique et de Géophysique, Université de Liège, Liège, Belgium
[11]Saint Petersburg State University, Atmospheric Physics Department, St. Petersburg, Russia
[12]Observatoire des Sciences de l'Univers Réunion (OSU-R), UMS 3365, Université de la Réunion, Saint-Denis, France
[13]National Institute for Environmental Studies (NIES), Tsukuba, Ibaraki 305-8506, Japan
[14]Graduate School of Environment Studies, Tohoku University, Sendai 980-8578, Japan
[15]Institute for Space-Earth Environmental Research (ISEE), Nagoya University, Nagoya, Japan
[16]Institute of Environmental Physics, University of Bremen, Bremen, Germany
[17]National Institute of Water and Atmospheric Research Ltd (NIWA), Lauder, New Zealand
[18]Karlsruhe Institute of Technology, IMK-IFU, Garmisch-Partenkirchen, Germany
[19]LERMA-IPSL, Sorbonne Université, CNRS, Observatoire de Paris, PSL Université, 75005 Paris, France
[20]Institute of Atmospheric Physics, Chinese Academy of Sciences (CAS), Beijing, China

*Correspondence to:* C. Vigouroux
(corinne.vigouroux@aeronomie.be)

**Abstract.** TROPOMI (the TROPOspheric Monitoring Instrument), on-board the Sentinel-5 Precursor satellite, has been monitoring the Earth's atmosphere since October 2017, with an unprecedented horizontal resolution (initially 7x3.5 km$^2$, upgraded to 5.5x3.5 km$^2$ since August 2019). Monitoring air quality is one of the main objectives of TROPOMI, with the measurements of important pollutants such as nitrogen dioxide, carbon monoxide, and formaldehyde (HCHO). In this paper we assess the quality of the latest HCHO TROPOMI products (version 1.1.[5-7]), using ground-based solar-absorption FTIR (Fourier Transform Infrared) measurements of HCHO from twenty-five stations around the world, including high, mid, and low latitude sites. Most of these stations are part of the Network for the Detection of Atmospheric Composition Change (NDACC), and they provide a wide range of observation conditions from very clean remote sites to those with high HCHO levels from anthropogenic

or biogenic emissions. The ground-based HCHO retrieval settings have been optimized and harmonized at all the stations, ensuring a consistent validation among the sites.

In this validation work, we first assess the accuracy of TROPOMI HCHO tropospheric columns, using the median of the relative differences between TROPOMI and FTIR ground-based data (BIAS). The pre-launch requirements of the TROPOMI HCHO accuracy are 40-80%. We observe that these requirements are well reached, with the BIAS found below 80% at all the sites, and below 40% at 20 of the 25 sites. The provided TROPOMI systematic uncertainties are well in agreement with the observed biases at most of the stations, except for the highest HCHO levels site where it is found to be underestimated. We find that, while the BIAS has no latitudinal dependence, it is dependent on the HCHO concentration levels: an overestimation (+26±5%) of TROPOMI is observed for very small HCHO levels ($<2.5\times10^{15}$ molec/cm$^2$), while an underestimation (-30.8%±1.4%) is found for high HCHO levels ($>8.0\times10^{15}$ molec/cm$^2$). This demonstrates the great value of such a harmonized network covering a wide range of concentration levels, the sites with high HCHO concentrations being crucial for the determination of the satellite bias at the regions of emissions, and the clean sites allowing a small TROPOMI offset to be determined. The wide range of sampled HCHO levels within the network allows the robust determination of the significant constant and proportional TROPOMI HCHO biases (TROPOMI=+ 1.10 (±0.05) $\times10^{15}$+ 0.64 (±0.03) × FTIR, in molec/cm$^2$).

Second, the precision of TROPOMI HCHO data is estimated by the median absolute deviation (MAD) of the relative differences between TROPOMI and FTIR ground-based data. The clean sites are especially useful to minimize a possible additional collocation error. The precision requirement of $1.2\times10^{16}$ molec/cm$^2$ for a single pixel is reached at most of the clean sites, where it is found that the TROPOMI precision can even be twice better ($0.5$-$0.8\times10^{15}$ molec/cm$^2$ for a single pixel). However, we find that the provided TROPOMI random uncertainties may be underestimated by a factor of 1.6 (for clean sites) to 2.3 (for high HCHO levels). The correlation is very good between TROPOMI and FTIR data (R=0.88 for 3 hours-mean coincidences; R=0.91 for monthly means coincidences). Using about 17 months of data (from May 2018 to September 2019), we show that the TROPOMI seasonal variability is in very good agreement at all of the FTIR sites.

The FTIR network demonstrates the very good quality of the TROPOMI HCHO products which is well within the pre-launch requirements for both accuracy and precision. This paper advises for a refinement of the TROPOMI random uncertainty budget and of the TROPOMI quality assurance values for a better filtering of the remaining outliers.

## 1 Introduction

TROPOMI (the TROPOspheric Monitoring Instrument), on-board the Sentinel-5 Precursor (S5P) satellite, has been monitoring the column amounts of atmospheric constituents since October 2017, at the unprecedented horizontal resolution of 7x3.5 km$^2$, upgraded to 5.5x3.5 km$^2$ since August 2019. This huge amount of data, delivered to the public and the scientific community, represents a big step to improve our knowledge of chemical and dynamical processes in the atmosphere. It is crucial to validate the quality of these new satellite data to trust and benefit their scientific exploitation. This paper focuses on the first quality assessment of the latest publicly available TROPOMI HCHO data products (v.1.1.[5-7]).

In the past, the HCHO satellite products have been validated at a few locations only, mainly using aircraft (Martin et al., 2004; Zhu et al., 2016, 2020), MAX-DOAS (Multi-AXis Differential Optical Absorption Spectroscopy) technique over land (Wittrock et al., 2006; De Smedt et al., 2015) or ship-based (Peters et al., 2012; Tan et al., 2018) and FTIR (Fourier Transform Infra-Red) technique (Jones et al., 2009; Vigouroux et al., 2009; De Smedt et al., 2015). However, given the high spatial heterogeneity of HCHO concentrations due to its short lifetime (a few hours), there is a crucial need for a more extended world coverage to assess unambiguously the satellites' achieved accuracy and precision. Furthermore, increasing the number of ground-based locations is not sufficient: it is also important to harmonize the reference data obtained at all the stations, in order to facilitate the interpretation of the satellite validation by minimizing the site-to-site biases. In this view, and in particular in the framework of the TROPOMI Calibration and Validation (Cal/Val) activities, we have developed HCHO retrieval settings that are suitable for any ground-based FTIR site, which have been consistently applied in Vigouroux et al. (2018) at twenty-one FTIR stations, most of them affiliated with NDACC (Network for the Detection of Atmospheric Composition Change). Vigouroux et al. (2018) described in detail the retrieval settings and the HCHO harmonized time-series obtained at these stations which cover a large range of HCHO concentrations, from very clean Arctic and oceanic sites to high HCHO levels sites, such as polluted cities (e.g. Paris or Mexico City) and sites close to large biogenic emissions like the Amazon basin (Porto Velho).

This paper presents the validation of the TROPOMI HCHO product (v.1.1.[5-7]) using an updated network of twenty-five ground-based FTIR stations. In the first section, the TROPOMI HCHO data are introduced with their uncertainty budget and their quality flag criteria. The second section describes the ground-based FTIR HCHO network and the characterization of these reference data (uncertainties and averaging kernels). Then, the validation procedure (collocation criteria, smoothing technique, definition of the quantities to be used in the quality assessment) is explained in Sect. 4. Finally, Sect. 5 shows the validation results using comparisons between TROPOMI and FTIR ground-based network data, leading to an assessment of the TROPOMI HCHO accuracy and precision, and the observed TROPOMI bias.

## 2 TROPOMI HCHO data

TROPOMI, on the S5P platform, is in a low-Earth afternoon polar orbit with a swath of 2600 km resulting in a daily global coverage (Veefkind et al., 2012). Operational Level 2 (L2) products include vertical columns of $O_3$, $SO_2$, $NO_2$, HCHO, CO and $CH_4$, as well as $O_3$ profile, aerosol layer height, cloud information and aerosol index. The spatial resolution, originally of $3.5x7$ $km^2$ has been increased to $3.5x5.5$ $km^2$ on 6 August 2019.

The prototype algorithm of the formaldehyde product is being developed at the Royal Belgian Institute for Space Aeronomy (BIRA-IASB) and the corresponding operational processor is being developed at the Remote Sensing Technology Institute (IMF) of the German Aerospace Center (DLR). The product has been declared operational and released to the public at the end of 2018. At the time of writing this paper, the latest product versions 1.1.[5-7] provide a consistent time series of Reprocessed+Offline (RPRO+OFFL) data, covering the period between May 2018 up to (at least) December 2019 (last access). The detailed validation results shown in Sect. 5 are obtained using this consistent time-series (RPRO+OFFL, from 2018-05-14

to 2019-12-31). The version numbers and their dates of change are given in Table 1, and further details are given in the Readme file [1]. The Near-Real-Time (NRTI) product, for the same versions 1.1.[5-7], started in December 2018 up to December 2019 (last access). This product has also been validated, but the results being very similar to the RPRO+OFFL validation, we do not show them in details in this paper.

**Table 1.** TROPOMI RPRO+OFFL complete time-series (versions 1.1.[5-7]) used in the present work.

| Date | Processor version | Relevant improvements (see Readme file [1]). |
|---|---|---|
| 2018-05-14 to 2018-11-28 | RPRO v.1.1.5 | Alignment of the configuration for NRTI, OFFL and RPRO chains regarding the Chemistry |
| 2018-11-28 to 2019-03-28 | OFFL v.1.1.5 | Transport Model input, leading to the same product quality. |
| 2019-03-28 to 2019-04-23 | OFFL v.1.1.6 | - Surface classification climatology updated |
| | | - Fixed a bug in the interpolation of the surface albedo climatology |
| | | - Fixed a problem regarding the retrieved CLOUD product parameters being too close to the |
| | | a-priori values. This might have affected the calculation of the HCHO in cloudy cases. |
| 2019-04-23 to 2019-12-31 | OFFL v.1.1.7 | No changes (for HCHO) with respect to previous version. |

The S5P HCHO retrieval algorithm is based on the DOAS method, and is directly inherited from the OMI QA4ECV product retrieval algorithm (https://doi.org/10.18758/71021031). It consists in a 3-step method (slant column retrieval, air mass factor calculation, and conversion to tropospheric column), fully described in De Smedt et al. (2018). The retrieval of the slant columns ($N_s$) is performed in the UV part of the spectra (in TROPOMI channel 3), in a fitting interval of 328.5-359 nm. The HCHO cross-section is from Meller and Moortgat (2000). Together with the HCHO cross-section, the absorptions of $NO_2$,

BrO, $O_3$ (at two temperatures) and $O_4$ are fitted. A Ring cross-section and two pseudo-cross sections to account for non-linear $O_3$ absorption effects are also included in the fit. References are given in De Smedt et al. (2018). All cross-sections have been pre-convolved for every row separately with an instrumental slit function adjusted just after launch. The DOAS reference spectrum is updated daily with an average of Earth radiances selected in the Equatorial Pacific region on the previous day. The result of the fit is therefore a differential slant column, showing increases over continental sources compared to the remote

background. The conversion from slant to tropospheric columns ($N_v$) is performed using a look up table of vertically resolved air mass factors ($M$) calculated at 340 nm with the radiative transfer model VLIDORT v2.6 (Spurr, 2008). Parameters for each ground pixel are the observation geometry, the surface elevation and reflectivity, including the clouds (that are treated as reflecting surfaces), and a priori tropospheric profiles. The surface albedo is taken from the monthly OMI albedo climatology (minimum Lambertian equivalent reflectivity, (Kleipool et al., 2008)) at the spatial resolution of $1°x1°$. A priori vertical pro-

files are specified using the TM5-MP daily forecast, at the same spatial resolution (Williams et al., 2017). Cloud properties are provided by the S5P operational product in its CRB mode (Cloud as Reflecting Boundary, Loyola et al. (2018)). A cloud correction based on the independent pixel approximation (Boersma et al., 2004) is applied for cloud fractions larger than 0.1. In order to correct for any remaining global offset and stripes, a background correction is applied based on HCHO slant columns from the 5 previous days in the Pacific Ocean ($N_{(s,0)}$), as described in De Smedt et al. (2018). Finally, the background vertical

---

[1] http://www.tropomi.eu/sites/default/files/files/publicSentinel-5P-Formaldehyde-Readme_20191213.pdf

column of HCHO, due to methane oxidation, is taken from the TM5 model in the reference region ($N_{(v,0)}^{CTM}$). The equation of the tropospheric HCHO vertical column can be written as follows:

$$N_v = \frac{(N_s - N_{(s,0)})}{M} + \frac{M_0}{M} \cdot N_{(v,0)}^{CTM}, \tag{1}$$

with $M_0$ the average of the air mass factors $M$ of the slant columns selected in the reference sector, the Pacific Ocean ($N_{(s,0)}$). Intermediate quantities and auxiliary data are all provided in the L2 files (http://www.tropomi.eu/sites/default/files/files/Sentinel-5P-Level-2-Product-User-Manual-Formaldehyde_v1.01.01_20180716.pdf).

Several diagnostic variables are provided together with the measurements. Quality assurance values (QA) are defined to perform a quick selection of the observations. QA>0.5 filters out most observations presenting an error flag or a solar zenith angle (SZA) larger than 70°, a cloud radiance fraction at 340 nm larger than 0.6 or an air mass factor smaller than 0.1. The product Readme file reports that in the current version, the QA values are not always correctly set over snow/ice regions or above 75° of SZA. They also need to be further checked over cloudy scenes. In the forthcoming S5P version 2, QA values will be refined, and will exclude data with surface albedo larger than 0.2 and snow/ice warning, and remaining SZA larger than 75°.

The tropospheric column uncertainty is divided into random (precision) and systematic components (accuracy), and is provided per pixel. It varies with the observation conditions. Over remote regions at moderate solar zenith angle, the precision of an individual observation is about $5 \times 10^{15}$ molec/cm$^2$. This value agrees with the standard deviation of the columns in the same region for a particular day. The random uncertainty is dominated by the random error on the slant columns. The tropospheric column accuracy is the combined systematic uncertainty resulting from the slant column, the air mass factor and the background correction errors. It varies between 30 and 60% of the columns. The column averaging kernel and the a priori profiles are provided for every observation.

## 3   Ground-based FTIR HCHO data

We show in Fig. 1 a map of the ground-based FTIR stations used in this TROPOMI validation. The background image represents the global TROPOMI monthly mean tropospheric columns for September 2018, illustrating the different HCHO levels sampled by the ground-based network: from clean Arctic and oceanic sites to very high-concentrations sites such as Porto Velho, in the Amazon basin.

Table 2 lists the ground-based FTIR stations, their coordinates and altitude, the spectrometer type, the retrieval code, and the team involved in the measurements and/or the retrievals of HCHO. For more details on the monitoring of FTIR solar absorption spectra at these stations, we refer to Vigouroux et al. (2018) and references therein, and, for the FTIR retrieval principles, to e.g. Vigouroux et al. (2009).

The same retrieval settings are used at all the stations to avoid introducing possible bias in the HCHO total columns between the stations and inconsistent comparisons with the satellite. Details are given in Vigouroux et al. (2018). The main settings that might be responsible for internal biases within the network are the spectroscopic database and the fitted spectral windows, the spectroscopic parameters being the main source of the FTIR HCHO systematic uncertainties. The HCHO spectral signatures

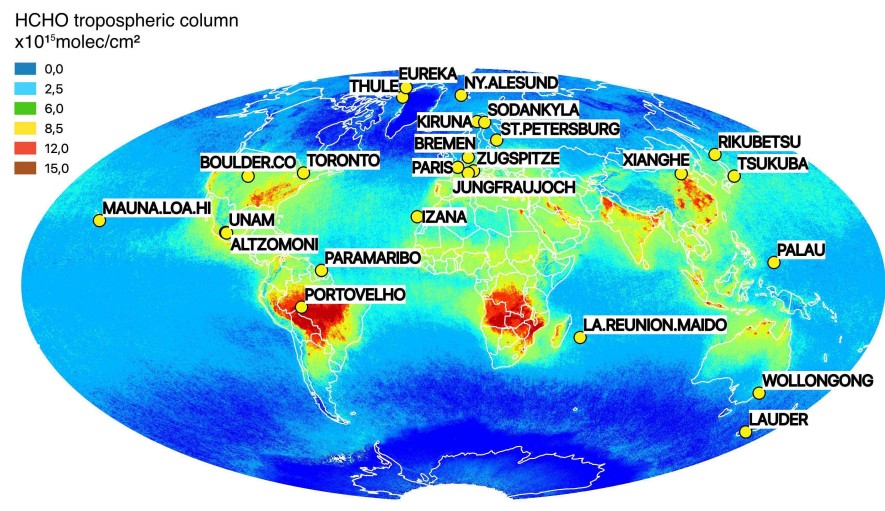

**Figure 1.** Network of ground-based FTIR stations providing HCHO total columns data. The background is the September 2018 monthly mean of TROPOMI HCHO tropospheric columns, averaged on a $0.2° \times 0.2°$ grid, using the HARP tool v.1.5 (https://atmospherictoolbox.org).

lie in the 3.6 $\mu$m region and belong to the $\nu_1$ and $\nu_5$ bands (fitted windows are: 2763.42 - 2764.17; 2765.65 - 2766.01 ; 2778.15 - 2779.1 ; 2780.65 - 2782.0, in cm$^{-1}$). The spectroscopic database used is the atm16 linelist by G. Toon (JPL), which can be found at http://mark4sun.jpl.nasa.gov/toon/linelist/linelist.html. This linelist is optimized for the main absorbing gases in the fitted windows (HDO, CH$_4$, O3, N$_2$O, CO$_2$) and is based on HITRAN 2012 (Rothman et al., 2013) for HCHO, which used the work of Jacquemart et al. (2010).

5    The retrieval codes used in the FTIR NDACC community are PROFITT9 (Hase et al., 2006) and SFIT4.0.9.4 (updated from SFIT2 (Pougatchev et al., 1995)), which are both based on the optimal estimation method (Rodgers, 2000). A past comparison exercise has shown a very good agreement between the retrieved products obtained with these two codes (Hase et al., 2004). Based on a priori profile information (from the model WACCM, Garcia et al. (2007)), and a L1 Tikhonov regularization matrix

10   (Tikhonov, 1963), low vertical resolution profiles can be retrieved in principle, as well as total columns. However, as described in Vigouroux et al. (2018), the degrees of freedom for signal are very low for HCHO (median value of 1.1 for all FTIR sites), meaning that we essentially have one piece of information. The FTIR total column averaging kernel shows a decrease of the sensitivity at the surface, which is quite similar to the TROPOMI sensitivity. This can be seen in Fig. 2, as an example for the Maïdo station. We also show in Fig. 2 the FTIR a priori profile at Maïdo, which is based on a climatology (1980-2020) from the

15   WACCM model calculated at Maïdo. A single profile is used for the whole time series at a specific station (Vigouroux et al., 2018), while TROPOMI uses daily a priori profiles from TM5 (Sect. 2). An example is shown in Fig. 2 for the 18th January 2019.

**Table 2.** FTIR stations that are contributing to the present work: location and altitude (in km a.s.l.), instrument type, retrieval code, team.

| Station | Latitude | Longitude | Altitude | Instrument | Code | Team |
|---|---|---|---|---|---|---|
| Eureka | 80.05° N | 86.42° W | 0.61 | Bruker 125 HR | SFIT4 | U. of Toronto |
| Ny-Ålesund | 78.92° N | 11.92° E | 0.02 | Bruker 120 HR | SFIT4 | U. of Bremen |
| Thule | 76.52° N | 68.77° W | 0.22 | Bruker 125 HR | SFIT4 | NCAR |
| Kiruna | 67.84° N | 20.40° E | 0.42 | Bruker 120/5 HR | PROFFIT | KIT–ASF ; IRF Kiruna |
| Sodankylä | 67.37° N | 26.63° E | 0.19 | Bruker 125 HR | SFIT4 | FMI ; BIRA |
| St-Petersburg | 59.88° N | 29.83° E | 0.02 | Bruker 125 HR | SFIT4 | SPbU |
| Bremen | 53.10° N | 8.85° E | 0.03 | Bruker 125 HR | SFIT4 | U. of Bremen |
| Paris | 48.97° N | 2.37° E | 0.06 | Bruker 125 HR | PROFFIT | Sorbonne U. |
| Zugspitze | 47.42° N | 10.98° E | 2.96 | Bruker 120/5 HR | PROFFIT | KIT–IFU |
| Jungfraujoch | 46.55° N | 7.98° E | 3.58 | Bruker 120 HR | SFIT4 | U. of Liège |
| Toronto | 43.60° N | 79.36° W | 0.17 | Bomem DA8 | SFIT4 | U. of Toronto |
| Rikubetsu | 43.46° N | 143.77° E | 0.38 | Bruker 120/5 HR | SFIT4 | Nagoya U. ; NIES |
| Boulder | 40.04° N | 105.24° W | 1.61 | Bruker 125 HR | SFIT4 | NCAR |
| Xianghe | 39.75° N | 116.96° E | 0.05 | Bruker 125 HR | SFIT4 | CAS ; BIRA |
| Tsukuba | 36.05° N | 140.12° E | 0.03 | Bruker 125 HR | SFIT4 | NIES ; Tohoku U. |
| Izaña | 28.30° N | 16.48° W | 2.37 | Bruker 120/5 HR | PROFFIT | AEMET ; KIT–ASF |
| Mauna Loa | 19.54° N | 155.57° W | 3.40 | Bruker 125 HR | SFIT4 | NCAR |
| Mexico City (UNAM) | 19.33° N | 99.18° W | 2.26 | Bruker Vertex 80 | PROFFIT | UNAM |
| Altzomoni | 19.12° N | 98.66° W | 3.98 | Bruker 120/5 HR | PROFFIT | UNAM |
| Palau | 7.34° N | 134.47° E | 0.03 | Bruker 120/5 M | SFIT4 | U. of Bremen |
| Paramaribo | 5.81° N | 55.21° W | 0.03 | Bruker 120/5 M | SFIT4 | U. of Bremen |
| Porto Velho | 8.77° S | 63.87° W | 0.09 | Bruker 125 M | SFIT4 | BIRA |
| Maïdo (LA.REUNION.MAIDO) | 21.08° S | 55.38° E | 2.16 | Bruker 125 HR | SFIT4 | BIRA |
| Wollongong | 34.41° S | 150.88° E | 0.03 | Bruker 125 HR | SFIT4 | U. of Wollongong |
| Lauder | 45.04° S | 169.68° E | 0.37 | Bruker 120 HR | SFIT4 | NIWA |

The FTIR uncertainty budget is calculated following the formalism of Rodgers (2000) and is described in Vigouroux et al. (2018). It is separated into random and systematic components. The random uncertainty is dominated at all sites by the measurement noise uncertainty, which can vary from site to site depending on the spectrometer. The uncertainty on the retrieved FTIR total columns for individual sites is given in Vigouroux et al. (2018) for the 21 sites involved at that time. We obtain a median random uncertainty of $2.3 \times 10^{14}$ molec/cm$^2$, with a large value of $11.1 \times 10^{14}$ molec/cm$^2$ only at Mexico City where a lower resolution instrument is used (Vertex 80). The smoothing uncertainty on the total column has a non negligible random component (median value of $1.2 \times 10^{14}$ molec/cm$^2$). With the inclusion of the smoothing error in the uncertainty budget, the

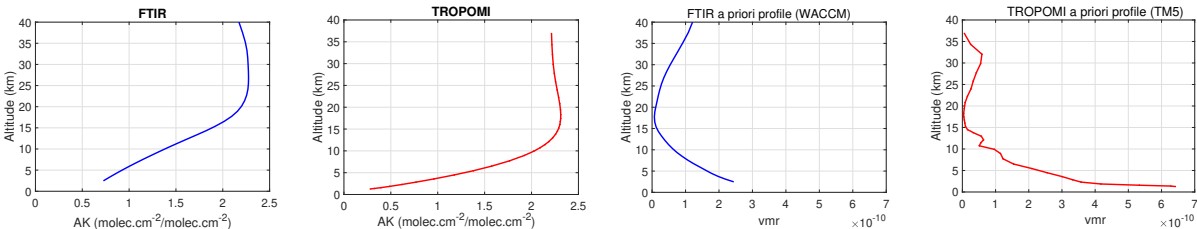

**Figure 2.** Left panels: typical total column averaging kernel (AK) from FTIR (blue) and TROPOMI (red) measurements at the Maïdo station (altitude: 2.2 km). Right panels: a priori profile used in the FTIR retrievals for the whole time series at Maïdo (blue), and an example of TROPOMI a priori profile at Maïdo (the 18th January 2019).

median total random uncertainty is 2.9 $\times 10^{14}$ molec/cm$^2$, which is very close to our empirical uncertainty estimation of 2.8 $\times 10^{14}$ molec/cm$^2$, based on the standard deviation of the differences between two FTIR individual subsequent measurements (within a maximum of 30 minutes interval), confirming our theoretical uncertainty calculation. Since the Vigouroux et al. (2018) paper, five more sites have joined the HCHO harmonized network. The mean random errors on individual FTIR mea-
surements are (in molec/cm$^2$): 1.4 $\times 10^{14}$, 2.7 $\times 10^{14}$, 2.2 $\times 10^{14}$, 5.2 $\times 10^{14}$, and 5.6 $\times 10^{14}$, for Jungfraujoch, Rikubetsu, Tsukuba, Palau, and Xianghe, respectively.

The forward model parameters median systematic uncertainty on the HCHO FTIR total columns is 13% in the network described by Vigouroux et al. (2018). As already mentioned, the dominating systematic uncertainty sources are the spectroscopic parameters: the line intensities and the pressure broadening coefficients of the fitted HCHO absorption lines. We use
10% for the three parameters: the line intensity, and the air- and self- broadening coefficients. The systematic uncertainty can be larger (up to 21-26%) at the stations using the PROFFIT9 retrieval code, due to an assumed uncertainty on the channeling that is not taken into account yet in the SFIT4 code. However, this channeling uncertainty can also be negligible at some sites (it depends on each instrument), and more investigation is needed at each station to avoid its under- or over-estimation. The median smoothing systematic uncertainty is 3.4%. For the five added sites, the median total systematic uncertainty is 13%
(Jungfraujoch, Tsukuba, Palau), or 14% (Rikubetsu, Xianghe), commensurate with the other sites.

## 4   Validation method

### 4.1   Collocation criteria

The precision of a single pixel TROPOMI HCHO measurement is expected to be below 1.2 $\times 10^{16}$ molec/cm$^2$ (pre-launch requirements) or even better, as 5 $\times 10^{15}$ molec/cm$^2$ for remote areas (after launch uncertainty analysis, see Sect. 2). These
values are quite large compared to the measured levels of HCHO (around 1.5 $\times 10^{15}$ molec/cm$^2$ for very clean sites to e.g. around 9 $\times 10^{15}$ molec/cm$^2$ for a city such Paris). It is therefore necessary to average several pixels in order to reduce the random uncertainty of the TROPOMI mean HCHO data, improve the detection level and increase the TROPOMI sensitivity to day-to-day variability. For this reason, we choose to average the TROPOMI pixels located within 20 km from the FTIR station.

Once we filter out the TROPOMI pixels that do not reach the recommended quality criteria (QA flag > 0.5; see Sect. 2), we obtain a median value of 34 pixels to average. In cloudy conditions, this number can be smaller. A collocation pair is kept when at least 10 pixels can be averaged. Higher number of pixels can be averaged for Arctic stations (around 45-60), which is useful due to the very low HCHO levels to be detected there. At sub-tropical / tropical stations, the median number of pixels

is around 20-29. The higher number of pixels in Arctic is due to the fact that each FTIR measurement is collocated to all S5P pixels that match the collocation criteria, even if these pixels originate from different orbits, with different overpass times. Before choosing the 20 km collocation criterion, we have tested several distances (10, 20, 30, 40, and 50 km). The 10 km criterion was discarded because of the poor number of remaining coincidences leading to less robust statistics. The 20 to 50 km criteria give similar biases between TROPOMI and FTIR. The standard deviations of the comparisons usually decrease slightly

with increasing collocation distance due to a smaller TROPOMI random uncertainty (more pixels to average), except at the most polluted sites. However, the ratio between the standard deviations and the random uncertainty budgets is increasing with the collocation distance at all sites, pointing to an increased random error due to the collocation. We therefore choose the 20 km distance to reduce the random spatial collocation error.

The time coincidence criterion is set to $\pm 3$ hours. This choice is a compromise to obtain significant number of coincidences

between TROPOMI and FTIR data, noting that the median FTIR measurement frequency is 5 per day (with a range of 3 to 10 depending on the station). A shorter time coincidence criterion decreases significantly the numbers of sampled collocated days and sometimes months, which is a limitation for checking the TROPOMI seasonality (sampled months: 267 for 1 h, and 305 for 3 h criteria). Note that a 6 h criterion would provide 20 additional sampled months: the critical stations are Mauna Loa, Altzomoni and Paramaribo, for which we would have coincidences back to May 2018. The standard deviations of the

TROPOMI / FTIR comparisons are usually smaller with a longer time coincidence criterion, but this can be explained by the increased number of pixels (improved TROPOMI precision on the mean) in the 6 h collocation, mainly at Arctic sites with increased number of multiple orbits. Despite the smaller standard deviations usually obtained within a 6 h criterion, we finally choose 3 h to reduce the possible impact of some passing plumes and of the HCHO diurnal cycle on the comparisons. The diurnal cycle at most of the FTIR stations can be found in Vigouroux et al. (2018) and its Supplement. At many stations no

significant diurnal cycle was observed but, in some cases, mainly polluted sites, we obtained a maximum around noon-1 p.m., close to the TROPOMI overpass time. At the Mexico City station, where the diurnal cycle amplitude is the greatest, the effect of collocation time (6 h vs 3 h) on the statistical bias is 4%.

## 4.2   Building inter-comparable products

Some manipulation of the original data products is needed before looking at the differences between TROPOMI and FTIR data.

Both measurements provide total columns (for FTIR) or tropospheric columns (for TROPOMI) that have a lower sensitivity near the ground (see Fig. 2), and their retrievals use a priori profile information that have been chosen differently (TROPOMI: daily a priori profiles from TM5; FTIR: single a priori profile from climatology of WACCM). To correct for this, for each S5P individual pixel collocated with each FTIR measurement, we use the comparison method described in Rodgers and Connor (2003). First, the a priori substitution is applied, using the S5P a priori profile $\boldsymbol{x}_{S,a}$ as the common a priori profile. For this,

the S5P a priori profile is regridded to the FTIR retrieval grid ($\boldsymbol{x}_{S,a/F}$) using a mass conservation algorithm (Langerock et al., 2015). In the rare situation where the satellite pixel elevation is above the FTIR site, the S5P a priori profile is extended to the FTIR instrument's altitude. The regridded S5P a priori $\boldsymbol{x}_{S,a/F}$ is then substituted following Rodgers and Connor (2003), and we finally use the corrected FTIR retrieved profile $\boldsymbol{x}'_F$ in the comparisons:

$$\boldsymbol{x}'_F = \boldsymbol{x}_F + (\mathbf{A}_F - \mathbf{I})(\boldsymbol{x}_{F,a} - \boldsymbol{x}_{S,a/F}), \tag{2}$$

where $\boldsymbol{x}_F$ is the original FTIR retrieved profile, $\mathbf{A}_F$ is the FTIR averaging kernel matrix, $\mathbf{I}$ is the unit matrix, and $\boldsymbol{x}_{F,a}$ is the FTIR a priori profile.

The next step, following Rodgers and Connor (2003), is to smooth the corrected FTIR profile with the S5P column averaging kernel $\boldsymbol{a}_S$. For that purpose we regrid the corrected FTIR profile $\boldsymbol{x}'_F$ to the S5P column averaging kernel grid ($\boldsymbol{x}'_{F/S}$) and apply
the smoothing equation:

$$c_F^{smoo} = c_{S,a} + \boldsymbol{a}_S(\boldsymbol{x}'_{F/S} - \boldsymbol{x}_{S,a}) \tag{3}$$

with $c_{S,a}$ the S5P a priori column derived from the S5P a priori profile. We obtain a smoothed FTIR column $c_F^{smoo}$ associated with a collocated TROPOMI pixel. In the case of mountain sites where the pixel altitude is below the instrument's height, the regridding of the FTIR profile $\boldsymbol{x}'_{F/S}$ is done such that the FTIR profile is extended with the S5P a priori profile (such an
extension is invariant under the latter smoothing equation). Note that this FTIR regridding to the satellite grid has also the advantage that only the FTIR profile up to the altitude of the satellite product (which is only a tropospheric column) remains in the regridded column: we therefore finally compare tropospheric columns in both products.

Next, we need to take into account that, for mountain stations, the difference between satellite columns and the original ground-based columns can be significant. To bring both satellite and smoothed FTIR column $c_F^{smoo}$ (which is calculated as
a column valid at the satellite's pixel surface) values to the scale of the original FTIR columns, we apply a scaling factor $f$ representative for the fraction of the partial column between the satellite pixel altitude and the FTIR station altitude. This scaling factor is derived from the satellite a priori profile and is defined as:

$$f = 1 - \frac{c_{S,a}^{\Delta z}}{c_{S,a}}, \tag{4}$$

where $c_{S,a}^{\Delta z}$ denotes the partial column derived from the S5P a priori profile between the pixel surface and the FTIR station.
The TROPOMI column $c_S$ and its random and systematic uncertainties are also scaled with the same factor, so that finally the collocated products are all expressed at the altitude of the FTIR site (and not of the pixel surface). For mountain stations, the scaling factor $f$, calculated for each satellite's pixel, can reach a minimum of 0.5 for stations located at about 2 km altitude from the satellite's pixel surface (Maïdo, Izaña, or Altzomoni), or even 0.3 at the higher sites Jungfraujoch and Zugspitze, while at sea-level sites it is of course close to 1.0. In the rare cases where the satellite pixel is above the FTIR station, we

apply the conversion factor $f = 1 + c_{S,a}^{\Delta z}/c_{S,a}$, where the satellite a priori profile is extrapolated to the station surface in order to calculate the partial column of the a priori between both altitudes.

The final step is to average the individual smoothed and scaled FTIR columns $c_F^{smoo} \times f$ that are taken within 3 h, and the TROPOMI $c_S \times f$ individual pixel columns that are available within 20 km (which can belong to different orbits), to form the collocated pair $\text{FTIR}_i$ and $\text{TROP}_i$ used in the next section.

## 4.3 Estimation of the TROPOMI accuracy and precision

In Sect. 5.1, we assess whether the TROPOMI accuracy is compliant with pre-launch requirements (40-80%, as reported in the ESA official document S5P-RS-ESA-SY-164, 2014, Table 3, p. 19). The accuracy of the TROPOMI HCHO measurements will be estimated by deriving the median of the relative differences (BIAS) between the collocated $\text{TROP}_i$ and the reference $\text{FTIR}_i$ data at each station:

$$\text{BIAS} = med(\frac{(\text{TROP}_i - \text{FTIR}_i)}{\text{FTIR}_i}). \tag{5}$$

We can note that the applied scaling factor $f$ (see previous section) does not affect the BIAS estimation, even at high mountains stations, because it cancels in the division.

For robust statistics, the median is preferred to the mean due to the presence of outliers (a few remaining TROPOMI outliers after the QA filter, and some very small FTIR values that give very large relative difference after the division in Eq. 5). The presence of TROPOMI outliers is minimized by using the median, but they should be ideally removed by the QA filter. An improvement of the QA value is foreseen in the next product version, which should improve, e.g., the filtering at Arctic sites (SZA$>75°$).

In the next section, we also compare the obtained BIAS with the systematic uncertainty on the difference $\sigma_{\text{syst}}$, to evaluate the TROPOMI uncertainty budget:

$$\sigma_{\text{syst}}^2 = (\sigma_{S,\text{syst}})^2 + \boldsymbol{a}_S^T \mathbf{S}_{F,\text{syst}} \boldsymbol{a}_S + \boldsymbol{a}_S^T (\mathbf{I} - \mathbf{A}_F) \mathbf{S}_{\text{var,syst}} (\mathbf{I} - \mathbf{A}_F)^T \boldsymbol{a}_S, \tag{6}$$

where $\sigma_{S,\text{syst}}$ is the systematic uncertainty of TROPOMI columns, as provided in the public release database (but scaled for altitude, see Sect. 4.2), $\boldsymbol{a}_S$ is the TROPOMI total column averaging kernel, and $\mathbf{S}_{F,\text{syst}}$ is the FTIR systematic covariance matrix provided in vmr$^2$ in the standardized GEOMS format converted in partial columns units. The last term is the impact of different low vertical resolution profile measurements (the smoothing error) on the comparisons (see Eq. 27 in Rodgers and Connor (2003)), where for the systematic uncertainty part, we account for possible bias on $\boldsymbol{x}_{S,a}$ by following von Clarmann (2014):
$\mathbf{S}_{\text{var,syst}} = (\boldsymbol{x}_{S,a} - <\boldsymbol{x}>)(\boldsymbol{x}_{S,a} - <\boldsymbol{x}>)^T$.
The $\boldsymbol{x}_{S,a} - <\mathbf{x}>$ is not known and we follow Vigouroux et al. (2018), with $\boldsymbol{x}_{S,a} - <\mathbf{x}> =$-50%, -20%, -10%, +10%, +8%, +5% for the ground-4 km; 4-8 km; 8-13 km; 13-25 km; 25-40 km; 40-120 km layers, respectively (expressed in molec/cm$^2$). The last term of Eq. 6 is found to be of the order of a few percent, therefore negligible in $\sigma_{\text{syst}}$. In practice, the systematic uncertainty on the difference $\sigma_{\text{syst}}$ is dominated by the TROPOMI systematic uncertainty of about 40%, FTIR having a median systematic uncertainty of only 13% with a maximum of 26% (See Sect. 3).

Similarly, the precision of the TROPOMI HCHO products is estimated in Sect. 5.2, not with the usual standard deviation which is not robust in case of outliers, but with the median absolute deviation (MAD, see Huber (1981)) of the differences (DIFF$_i$=TROP$_i$-FTIR$_i$):

$$\text{MAD} = k \times med(abs(\text{DIFF}_i - med(\text{DIFF}_i))), \tag{7}$$

where $k = 1.4826$ for a correspondence with the 1-$\sigma$ standard deviation for normal distribution without outliers.

In Sect. 5.2, we compare the obtained MAD to the random uncertainty on the differences $\sigma_{\text{rand}}$, which is calculated following Rodgers and Connor (2003):

$$\sigma_{\text{rand}}^2 = (\sigma_{S,\text{rand}})^2 + \boldsymbol{a}_S^T \mathbf{S}_{F,\text{rand}} \boldsymbol{a}_S + \boldsymbol{a}_S^T (\mathbf{I} - \mathbf{A}_F) \mathbf{S}_{\text{var,rand}} (\mathbf{I} - \mathbf{A}_F)^T \boldsymbol{a}_S, \tag{8}$$

where where $\sigma_{S,\text{rand}}$ is the random uncertainty of TROPOMI columns, as provided in the public release database (but scaled for altitude, see Sect. 4.2), $\mathbf{S}_{F,\text{rand}}$ is the FTIR random covariance matrix, and $\mathbf{S}_{\text{var,rand}}$, to take into account the impact of low vertical resolution in the random part of the uncertainty, is the natural variability matrix chosen to be 50%, 50%, 40%, 35%, 30%, 30%, 10% for the ground-4 km; 4-8 km; 8-13 km; 13-25 km; 25-40 km; 40-120 km layers, respectively (expressed in molec/cm$^2$). As for the systematic uncertainty part, the random uncertainty on the difference is dominated by the TROPOMI random uncertainty (median of about 1.1 $\times 10^{15}$molec/cm$^2$ for TROP$_i$ within 20 km), while FTIR$_i$ has a median random uncertainty of 2.0 $\times 10^{14}$molec/cm$^2$. The last term of Eq. 8 is comparable to the FTIR one (median value of 2.4 $\times 10^{14}$molec/cm$^2$).

We can use MAD as an upper limit of the TROPOMI precision, since collocation in space and time of the sounded air-masses are never perfect. It is compared in the next section to the pre-launch precision requirement. The MAD estimation is influenced by the scaling factor $f$, which is important only for high altitude sites (Sect. 4.2). It should be interpreted as an estimation of the precision of a TROPOMI column that would be measured at the altitude of the FTIR site. The random uncertainty on the differences are also expressed at the altitude of the FTIR site, so that the comparison between MAD and $\sigma_{\text{rand}}$ is always valid.

The observed BIAS between TROPOMI and the reference FTIR data is statistically significant if it exceeds its statistical error: ERR$_B = 2 \times \text{MAD}/\sqrt{n}$ (with $n$ the number of coincidences).

## 5 Validation results

In this section, we provide a table and plots for the offline (RPRO+OFFL) HCHO TROPOMI product. We do not show detailed results for the near real time (NRTI) product (version 1.1.[5-7]) because they are very similar to the offline version. Numbers for the main conclusions will be given in the text for this NRTI product.

### 5.1 TROPOMI observed BIAS and accuracy

In Table 3, we provide, at each individual FTIR station, the mean of the FTIR HCHO total columns (mean FTIR), the obtained median of the relative differences BIAS (in % to compare with the pre-launch TROPOMI accuracy requirements of 40-80%,

Eq. 5), the error on the BIAS ($ERR_B$), and the number of collocated pairs $n$. The systematic uncertainty on a single difference is also given (in %, calculated from Eq. 6 where each term has been expressed in %, dividing by each instrument individual HCHO column).

We have ordered the stations, not in decreasing latitudes as in Table 2, but in increasing mean HCHO FTIR columns. The reason is that we observe a tendency of the BIAS between TROPOMI and FTIR: while the BIAS is always (with the exception of Eureka) positive or not significant (if BIAS<$ERR_B$) for very clean to clean sites with HCHO mean levels lower than $6.5 \times 10^{15}$ molec/cm$^2$, it is negative and very consistent for the stations with higher HCHO levels, ranging from 8.7 to $28.6 \times 10^{15}$ molec/cm$^2$ (-29 to -36 %) with small error on the bias (2 to 6 %). Note that the BIAS is also consistent at Paramaribo (-26%) but with larger error (14%), due to small number of collocations. This dependence of the TROPOMI bias on the HCHO concentration levels can be visualized in Fig. 3, where the BIAS at each station is plotted as a function of the mean FTIR columns. It is therefore not appropriate to use the median bias obtained using the data from all stations together (-10%), if one wants to correct the TROPOMI HCHO data in model inversion studies. If we calculate the median of the differences for HCHO FTIR columns >$8.0 \times 10^{15}$ molec/cm$^2$, we obtain a significant negative bias of -30.8$\pm$1.4%. The detection of this bias is especially important for modeling studies that use satellite data to optimize the volatile organic compound emissions sources, as done in e.g. Fortems-Cheiney et al. (2012); Stavrakou et al. (2015) with OMI and GOME-2. The bias for clean HCHO levels (<$2.5 \times 10^{15}$ molec/cm$^2$) is significantly positive (+26 $\pm$5%).

The validation results for the NRTI TROPOMI products give very similar results: a negative BIAS (-31.7$\pm$1.8%) for the high HCHO levels (>$8.0 \times 10^{15}$ molec/cm$^2$) and a positive one (+22 $\pm$7%) for low HCHO levels (<$2.5 \times 10^{15}$ molec/cm$^2$). The small differences are mainly due to the different sampling of the comparisons (NRTI data are since December 2018, while the OFFL data are since May 2018).

The different TROPOMI BIAS at different HCHO levels is pointing to the presence of two kinds of bias: a constant one and a proportional one. They can be obtained by using the scatter plot of the two instruments, shown in Fig. 4: the constant bias is the intercept of the linear relationship between TROPOMI and FTIR, while the proportional bias is given by its slope. But this has to be done carefully: a usual linear regression by ordinary least squares (OLS) is not statistically robust and can give spurious results in the presence of outliers and/or heteroscedasticity. We are confronted to both problems in our scatter plot: we do have outliers and the uncertainty is increasing with HCHO levels. Therefore, we use the robust Theil-Sen estimator (Sen, 1968) where the slope $s$ of the scatter plot is the median of the slopes of the lines through all pairs of data points ($TROP_j$ - $TROP_i$)/($FTIR_j$ - $FTIR_i$), with $FTIR_j \neq FTIR_i$. The intercept $b$ is then the median of ($TROP_i - s \times FTIR_i$). Using this robust estimator, we obtain the relation: TROP = 0.64 $\times$ FTIR + $1.10 \times 10^{15}$ molec/cm$^2$. We have calculated the uncertainties in $s$ and $b$ using $2 \times MAD/\sqrt{n}$, with MAD the median absolute deviation of the slopes and intercepts of the pairs of data points, and $n$ the numbers of pairs. We obtain an uncertainty of 0.03 and $0.05 \times 10^{15}$ molec/cm$^2$ for $s$ and $b$ respectively. Therefore, both the constant ($1.10 \pm 0.05 \times 10^{15}$ molec/cm$^2$) and proportional (0.64$\pm$0.03%) biases are significant.

Using the scatter plot to derive the constant and proportional biases is very sensitive to the range of observed values. As an example, if one would only use HCHO FTIR data >$8.5 \times 10^{15}$ molec/cm$^2$, one would obtain a slope of 0.51 and an intercept of $3.2 \times 10^{15}$ molec/cm$^2$), which would point to a strong overestimation and underestimation of the constant and proportional

**Table 3.** Validation of TROPOMI RPRO+OFFL. Please note that the ordering of the sites is by increasing mean HCHO column. For each station: mean of the HCHO FTIR total columns (in molec/cm$^2$), median of the relative differences BIAS=$med((\text{TROP}_i\text{-FTIR}_i)/\text{FTIR}_i)$ and its error ERR$_\text{B}$ (in %, see text), number of collocated pairs $n$, systematic uncertainty on a single difference $\sigma_\text{syst}$ (in %, Eq. 6), median absolute deviation (MAD, in molec/cm$^2$, Eq. 7), random uncertainty on a single difference $\sigma_\text{rand}$ (in molec/cm$^2$, Eq. 8), and pre-launch TROPOMI precision requirements associated to the choice of 20 km around the station Requ=$1.2\times10^{16}/\sqrt{n}_\text{pix}$ molec/cm$^2$, with $n_\text{pix}$ the mean number of pixels averaged in the collocated TROPOMI data. The Pearson correlation coefficient $R$ is given for individual coincidences ($\pm3$ h) and for monthly means of coincident data.

| Station | mean FTIR molec/cm$^2$ | BIAS $\pm$ ERR$_\text{B}$ % | $n$ | $\sigma_\text{syst}$ % | MAD molec/cm$^2$ | $\sigma_\text{rand}$ molec/cm$^2$ | Requ. molec/cm$^2$ | $n_{pix}$ | $R$ indiv. | $R$ monthly |
|---|---|---|---|---|---|---|---|---|---|---|
| Jungfraujoch | 1.24E+15 | 19 $\pm$ 15 | 87 | 58 | 9.0E14 | 5.6E14 | 2.5E15 | 24 | 0.61 | 0.70 |
| Zugspitze | 1.36E+15 | 52 $\pm$ 10 | 184 | 59 | 7.8E14 | 5.0E14 | 2.1E15 | 33 | 0.71 | 0.86 |
| Mauna Loa | 1.60E+15 | 52 $\pm$ 22 | 52 | 54 | 1.3E15 | 8.8E14 | 2.5E15 | 23 | -0.09 | -0.05 |
| Eureka | 1.65E+15 | -40 $\pm$ 11 | 114 | 97 | 1.1E15 | 5.3E14 | 1.8E15 | 45 | 0.22 | 0.43 |
| Maïdo | 1.86E+15 | 5 $\pm$ 9 | 155 | 43 | 1.0E15 | 7.1E14 | 3.0E15 | 16 | 0.45 | 0.53 |
| Ny-Ålesund | 1.90E+15 | 43 $\pm$ 20 | 47 | 41 | 1.2E15 | 4.9E14 | 1.7E15 | 52 | 0.35 | 0.38 |
| Thule | 2.06E+15 | -3 $\pm$ 5 | 346 | 57 | 9.6E14 | 4.8E14 | 1.6E15 | 60 | 0.56 | 0.82 |
| Izaña | 2.07E+15 | 13 $\pm$ 10 | 97 | 83 | 8.9E14 | 6.4E14 | 2.5E15 | 24 | 0.47 | 0.79 |
| Altzomoni | 2.44E+15 | 66 $\pm$ 18 | 67 | 42 | 1.6E15 | 8.9E14 | 2.6E15 | 22 | 0.50 | 0.86 |
| Kiruna | 2.44E+15 | 50 $\pm$ 12 | 146 | 67 | 1.5E15 | 8.8E14 | 1.7E15 | 60 | 0.64 | 0.72 |
| Lauder | 2.54E+15 | -11 $\pm$14 | 225 | 78 | 2.6E15 | 1.3E15 | 2.1E15 | 33 | 0.38 | 0.65 |
| Rikubetsu | 3.16E+15 | 26 $\pm$ 40 | 16 | 50 | 2.8E15 | 1.0E15 | 1.9E15 | 41 | 0.45 | 0.60 |
| Palau | 3.80E+15 | 0 $\pm$15 | 10 | 36 | 9.8E14 | 8.2E14 | 2.7E15 | 20 | 0.15 | 0.33 |
| Sodankyla | 4.15E+15 | 8 $\pm$ 7 | 307 | 51 | 2.5E15 | 1.2E15 | 1.7E15 | 48 | 0.51 | 0.69 |
| Boulder | 5.91E+15 | -1 $\pm$ 9 | 103 | 50 | 2.2E15 | 1.3E15 | 2.2E15 | 31 | 0.79 | 0.90 |
| St-Petersburg | 6.21E+15 | -4 $\pm$ 8 | 158 | 44 | 3.0E15 | 1.2E15 | 1.9E15 | 42 | 0.68 | 0.78 |
| Wollongong | 6.36E+15 | 9 $\pm$ 8 | 322 | 54 | 3.3E15 | 1.9E15 | 2.3E15 | 27 | 0.78 | 0.94 |
| Tsukuba | 7.05E+15 | -23 $\pm$12 | 34 | 44 | 3.1E15 | 1.2E15 | 2.2E15 | 31 | 0.68 | 0.51 |
| Bremen | 7.77E+15 | -5 $\pm$ 12 | 46 | 39 | 3.2E15 | 1.4E15 | 1.8E15 | 43 | 0.63 | 0.68 |
| Paramaribo | 8.43E+15 | -26 $\pm$14 | 15 | 36 | 3.3E15 | 1.3E15 | 2.5E15 | 23 | 0.12 | 0.14 |
| Paris | 8.72E+15 | -29 $\pm$6 | 128 | 44 | 3.1E15 | 1.2E15 | 1.9E15 | 41 | 0.76 | 0.79 |
| Toronto | 1.06E+16 | -34 $\pm$4 | 251 | 38 | 3.7E15 | 1.6E15 | 2.3E15 | 29 | 0.73 | 0.95 |
| Xianghe | 1.43E+16 | -33 $\pm$2 | 384 | 38 | 5.5E15 | 2.1E15 | 2.0E15 | 36 | 0.86 | 0.97 |
| Mexico City (UNAM) | 1.92E+16 | -27 $\pm$ 4 | 154 | 32 | 5.9E15 | 2.6E15 | 2.5E15 | 25 | 0.34 | 0.27 |
| Porto Velho | 2.86E+16 | -36 $\pm$ 3 | 81 | 31 | 8.3E15 | 3.6E15 | 2.2E15 | 29 | 0.81 | 1.00 |
| All stations BIAS | 6.60E+15 | -10 $\pm$2 | 3529 | 48 | 2.4E15 | 1.2E15 | 2.1E15 | 34 | 0.87 | 0.91 |
| Low HCHO BIAS FTIR<$2.5\times10^{15}$ | 1.64E+15 | +26 $\pm$5 | 1321 | 52 | 1.3E15 | 7.7E14 | 2.2E15 | 31 | 0.40 | |
| High HCHO BIAS FTIR>$8.0\times10^{15}$ | 1.57E+16 | -30.8 $\pm$1.4 | 952 | 46 | 4.8E15 | 2.1E15 | 2.1E15 | 33 | 0.88 | |

biases, respectively. This would lead to erroneous prediction of TROPOMI overall bias outside the range of observed values. This demonstrates why such a ground-based network, covering very clean sites to high HCHO level sites, is crucial to provide a good estimate of both constant and proportional biases of TROPOMI.

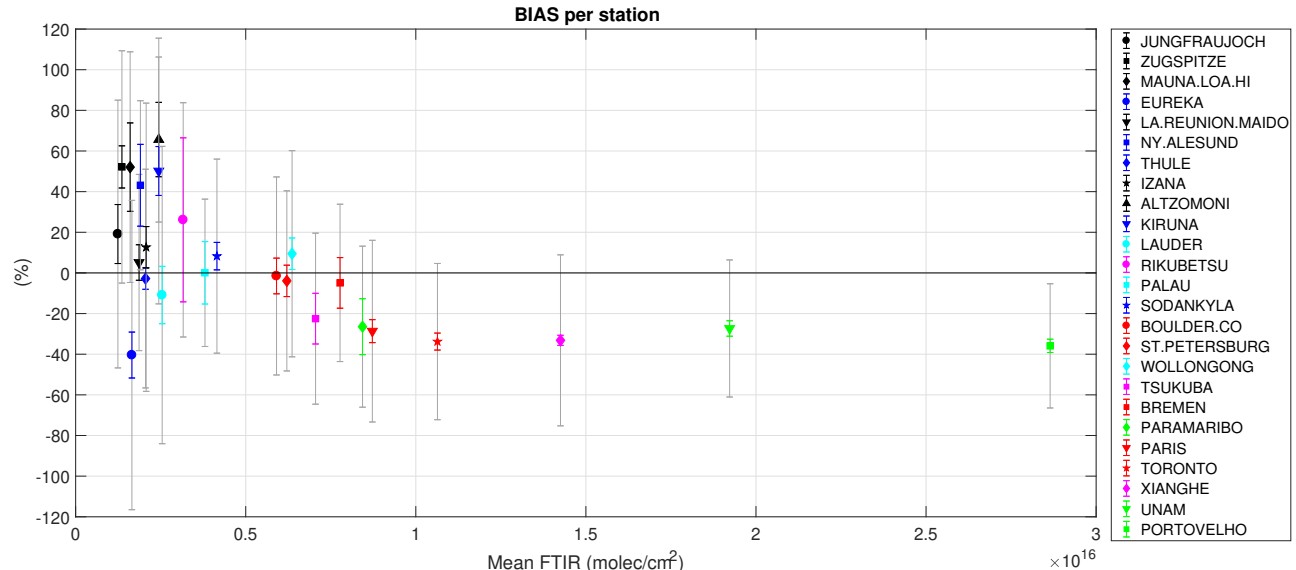

**Figure 3.** BIAS at each station (in %) as a function of the mean FTIR total columns (molec/cm$^2$). The gray bars are the systematic uncertainty on the differences $\sigma_{syst}$, and the colored error bars are the 2-$\sigma$ error on the bias ERR$_B$. Black markers are for mountains stations, blue for Arctic stations, cyan for Oceania/Australia/New Zealand, magenta for China/Japan, red for mid-latitude Europe/North America, and green for Central/South America.

The BIAS given in Table 3 are a combination of the constant and proportional biases, and can be use to statistically assess the TROPOMI HCHO overall accuracy. We can easily see from Table 3 that all BIAS values are within the upper limit of the pre-launch requirement of 80%, and they are within the 40% requirement lower limit for 20 of the 25 stations. The five stations exceeding a 40% BIAS are clean (Arctic or mountains) sites, with mean HCHO columns below $2.5 \times 10^{15}$ molec/cm$^2$. But these are sites where the systematic uncertainty on the differences (see Table 3 and Eq. 6) are usually also the largest, leading to a good correspondence between observed higher BIAS and higher calculated uncertainty for 3 of these 5 stations (Zugspitze, Mauna Loa, and Kiruna).

Therefore, we can conclude that the TROPOMI HCHO accuracy satisfies the pre-launch requirements and that the systematic uncertainty budget is in very good agreement with observed bias, except at a very few stations (Ny-Ålesund 43>41%, Altzomoni 71>42%, and Porto Velho 36>31%). At most of the other stations, the reported systematic uncertainty tends to be larger than the BIAS. We find the same conclusions on TROPOMI accuracy when making comparisons with the NRTI products.

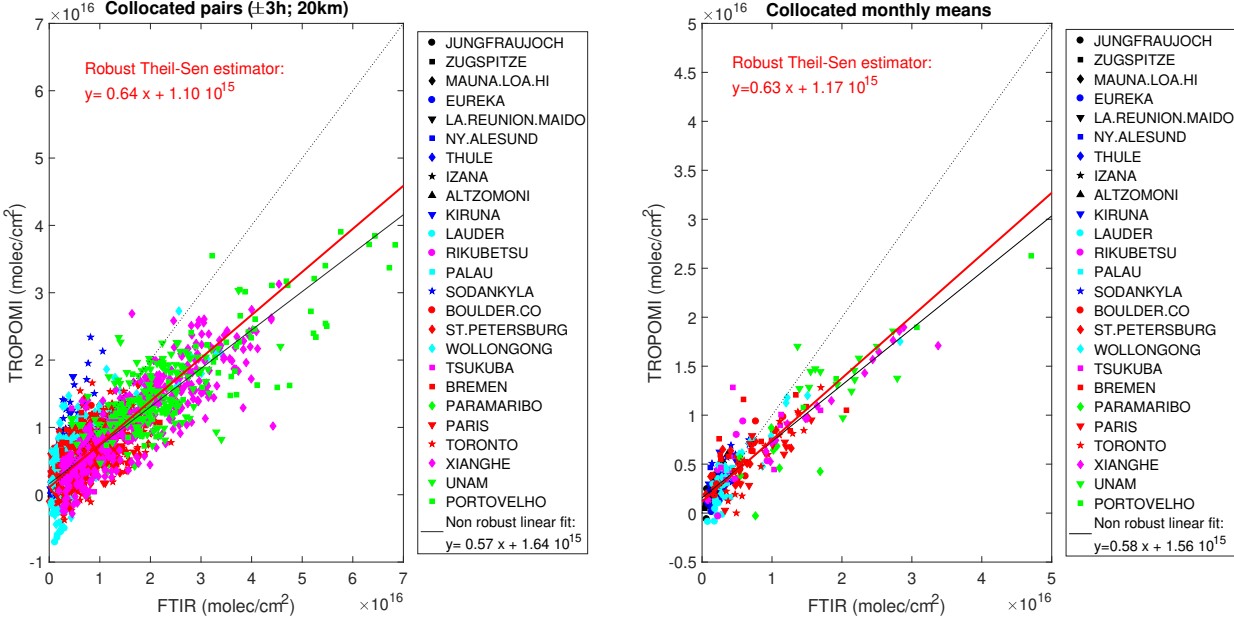

**Figure 4.** Scatter plots of TROPOMI versus FTIR data, for individual collocated pairs ($\pm 3$ h; left panel), and for the monthly means of collocated data (right panel). The non robust OLS fit of the data is given in the legend and plotted as a black line, while the slope and intercept obtained with robust Theil-Sen estimator is given by the red line and text.

The systematic uncertainties leading to the observed constant and proportional biases of our study have been calculated as described in Sect. 3 of De Smedt et al. (2018). From the error propagation of the HCHO TROPOMI tropospheric columns (see Eq. 1), it can be found that the proportional bias is more likely due to air mass factor ($M$) uncertainties $\sigma_M$, while the constant bias is more likely due to the uncertainties of the slant columns uncertainties $\sigma_{N,S}$ and to the uncertainty of the background correction of the slant columns. This can be seen in Eq. 13 of De Smedt et al. (2018), where $\sigma_M$ is proportional to $N_s - N_{(s,0)}$. We can list some known difficulties of the satellite product:

- The negative bias over high HCHO levels sites (biomass burning or mega-cities) could be due to aerosol effects. There is no plan to include a correction for aerosols in the operational product, but specific studies are foreseen to check its impact in a scientific product.

- The positive bias over clean polar sites could be due to the solar zenith angle (SZA) dependency of the slant columns fit results (because of spectral interferences with ozone and BrO). As explained in Sect. 2, the QA values need to be improved at large SZA, which is foreseen in the next version.

- The current albedo climatology is too coarse for TROPOMI, which could be especially a problem for polar, mountain or coastal sites. A climatology based on TROPOMI measurements is under development.

– It is also foreseen to test a regional model at higher spatial resolution for an improvement of the a priori HCHO profiles. This should improve the TROPOMI retrieved product, especially at polluted sites. However, the validation presented here is already taking the a priori information and averaging kernels into account. We therefore do not expect an important effect of the improved a priori profiles on the validation results.

## 5.2 Observed TROPOMI precision

For discussing the observed TROPOMI precision, we provide in Table 3 the MAD for each station (in absolute value to compare with the pre-launch precision requirement of $1.2 \times 10^{16}$ molec/cm$^2$ for a single pixel, Eq. 7). Indeed, for each site, the MAD is an upper limit for the TROPOMI precision as determined by our validation (see Sect. 4.3), while the $\sigma_{\mathrm{rand}}$ given in Table 3 is an approximation of the precision as provided in the satellite product (because the FTIR random uncertainty is much smaller than the TROPOMI's one). The detection limit, usually defined as being three times the precision, can then be obtained at each station for an average of TROPOMI pixels within 20 km, by multiplying by three either the $\sigma_{\mathrm{rand}}$ (theoretical estimation by TROPOMI data providers, which is probably underestimated, as seen below), or the MAD (upper limit determined by the validation), both given in Table 3. The precision pre-launch requirement is provided at each site taking into account the mean number of pixels n$_{pix}$ involved in the collocated TROPOMI data (Requ.=$1.2 \times 10^{16}$ molec/cm$^2/\sqrt{n}_{pix}$). We see that for all the cleanest sites ($<2.5 \times 10^{15}$ molec/cm$^2$), where an additional collocation uncertainty is expected to be small, the MAD is well within the pre-launch requirements. The MAD for these cleanest sites has a median of $1.3 \times 10^{15}$ molec/cm$^2$, and a minimum of $0.9 \times 10^{15}$ molec/cm$^2$. This is a good estimate of the precision that TROPOMI can reach in remote conditions. For a single pixel, the TROPOMI best precision at remote conditions is therefore $5$-$8 \times 10^{15}$ molec/cm$^2$.

It must be noted that the pre-launch HCHO precision requirements were chosen based on pre-launch requirements for the instrument signal to noise ratio (equivalent to OMI). The actual signal to noise of the measurements appears to be better than the requirements, especially in the HCHO wavelength fitting range. Furthermore, the good quality of the recorded spectra allowed to increase the size of the TROPOMI HCHO fitting spectral interval just after launch, further improving the precision of the slant columns. Indeed, as seen in Table 3, only at the three highest HCHO levels sites (Xianghe, Mexico City, and Porto Velho) the provided random uncertainties are as high as the pre-launch requirements. The actual provided random uncertainty are smaller, and we can see that, even for clean sites, the observed MAD is larger than the random uncertainty on the differences by a factor of 1.6. This factor increases up to 1.8 if we take into account all the stations, but this is expected due to a collocation uncertainty that should have more impact at high-levels sites (the factor rises up to 2.3 for high HCHO levels $>8.0 \times 10^{15}$ molec/cm$^2$). Our comparisons suggest that the TROPOMI random uncertainty is underestimated by at least a factor of 1.6 and up to maximum of 2.3 (if one would assume the collocation uncertainty to be smaller than the TROPOMI uncertainty). This underestimation could be due to the fact that currently the uncertainties associated to the air mass factor calculation and to the background correction step are assumed to be fully systematic. The discrimination between random and systematic part of the uncertainties might be refined in the future, based on such validation results.

## 5.3 Observed TROPOMI monthly variability

The Pearson correlation coefficient is very good for the collocated monthly means of TROPOMI and FTIR data (0.91, see Table 3 and Fig. 4), and is usually good for individual sites. However, Pearson correlation is not robust and can give a wrong conclusion when only few data are coincident, especially when outliers are present. We have 17 months of coincident TROPOMI and FTIR measurements in the best cases, while only 4 for the newest stations Palau and Porto Velho. We therefore verify that the TROPOMI precision allows the seasonal variability to be well captured, even at very clean sites which can be at the limit of the satellite detection, by plotting the individual monthly mean time-series in Fig. 5.

The seasonal variability, with a maximum in July-August, is well observed at all the Arctic sites (Eureka, Ny-Ålesund, Thule, Kiruna, and Sodankylä). The monthly mean correlation is better than 0.69, except at Eureka and Ny-Ålesund. It can be seen in Fig. 5 that Sept. 2019 is very high in TROPOMI data at Ny-Ålesund, and only 1 coincidence is found for this month. Removing this last outlier gives a 0.76 correlation coefficient at this station. The northern mid-latitude clean sites (mountains: Jungfraujoch, Zugspitze, Izaña) also display a seasonal variability in very good agreement, with correlation coefficients higher than 0.70. The Japanese clean site Rikubetsu shows poorer correlation (0.60) but only few data are in coincidence. The stations where we find the poorer correlations are the oceanic sites. The poorest one is Mauna Loa, but this is expected due to the very small seasonal variability there, associated to a small number of coincidences. A similar situation is observed at the other recent oceanic site Palau, where only 4 months of data are available. At the oceanic site Maïdo, we find a good agreement in most of the months but not in October-December, which are the predominant biomass burning months in the region so the collocation of the plumes might play a role there. Finally at Lauder, TROPOMI shows many negative values in the beginning of the period (May-Sept. 2018), which is responsible for a lower correlation (0.65) and for the negative bias there (although not significant), while other clean sites show usually positive ones (see Table 3).

The higher HCHO level sites show a TROPOMI seasonal variability in very good agreement with FTIR, with correlation larger than 0.90 for Boulder, Wollongong, Toronto, Xianghe, and Porto Velho. At Tsukuba, removing the outlier of 1 coincidence in November 2018 increases the correlation to 0.93. The poorest correlation (0.14) is found at the coastal site Paramaribo, where usually only one coincidence per month is found. Looking at the highest HCHO level sites, these monthly mean time-series also confirm that TROPOMI has more difficulty to reproduce the months with the highest enhancements, which is responsible for the significant negative bias (-31%) found in the previous section for high HCHO levels ($>8.0\times10^{15}$ molec/cm$^2$).

## 6 Conclusions

We have used a network of twenty-five FTIR stations, most of them affiliated to NDACC, to validate the latest TROPOMI HCHO tropospheric columns (v.1.1.[5-7]). This network covers a wide range of concentrations, from very clean Arctic, oceanic and mountain sites, with columns that can be lower than $10^{14}$ molec/cm$^2$, to high HCHO level sites such Mexico city or Porto Velho, near the Amazonian forest, where columns up to $7\times10^{16}$ molec/cm$^2$ have been observed.

We found an overestimation (+26±5%) of TROPOMI OFFL products for very small HCHO columns ($<2.5\times10^{15}$ molec/cm$^2$) and an underestimation of TROPOMI of about -30.8% (±1.4%) for high HCHO columns ($>8.0\times10^{15}$ molec/cm$^2$), which can

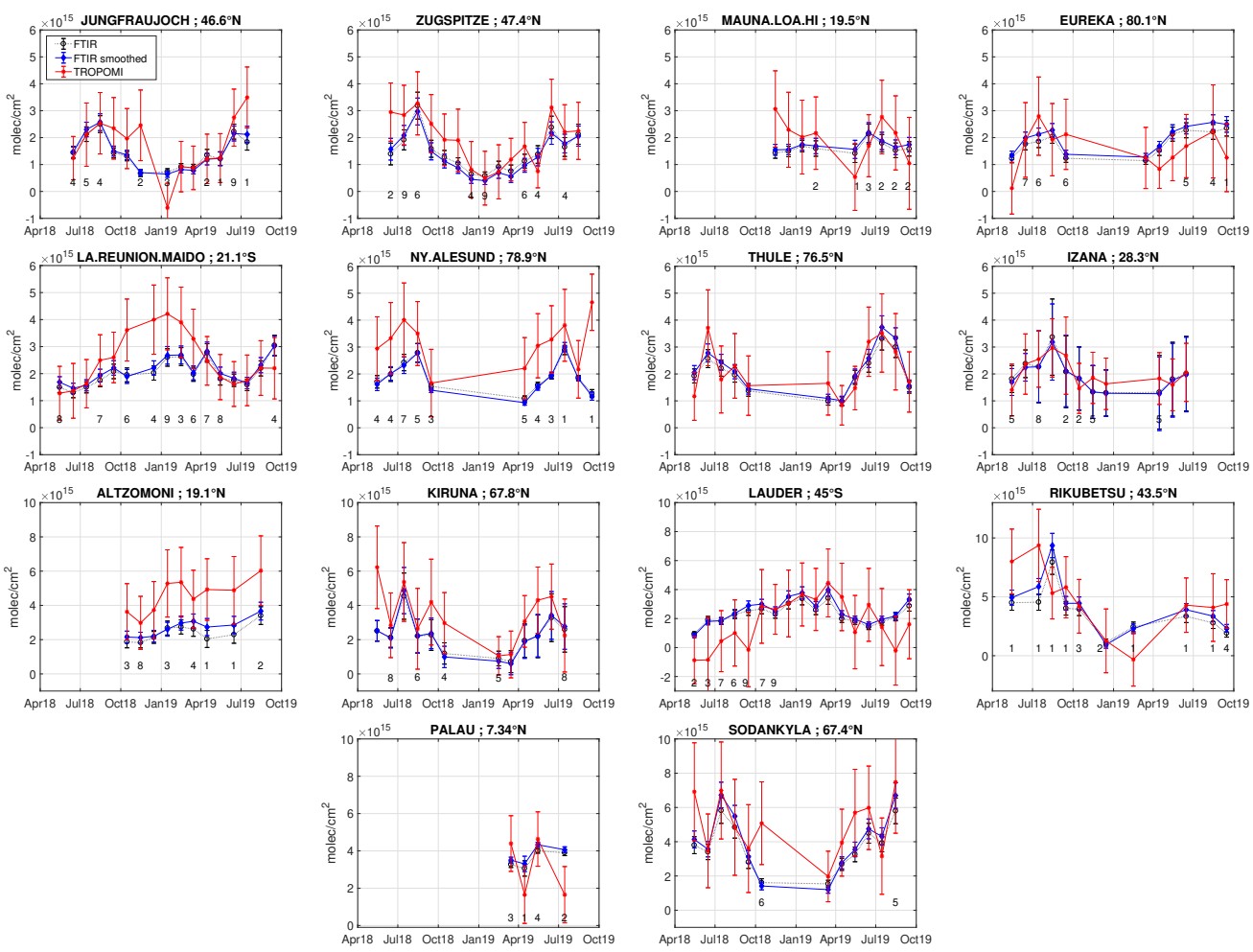

**Figure 5.** Monthly means time-series of FTIR raw data (black), FTIR data smoothed with the TROPOMI column averaging kernel (blue), and TROPOMI (red) at each site. Only data in coincidences are included in the monthly mean to avoid sampling bias. When the number of coincidences within one month is smaller than 10, it is written below the monthly mean.

be used, e.g., to correct TROPOMI data near emissions sources. The results are very similar for NRTI products (+22 ±7% and -31.7±1.8% for small and high columns, respectively), and the differences are mainly due to the different period of available TROPOMI v.1.1.[5-7] products. Our wide range of HCHO levels and the use of the Theil-Sen method allow us to derive robust and significant constant (intercept) and proportional (slope) biases of TROPOMI (TROP=+ 1.10 (±0.05) ×$10^{15}$+ 0.64 (±0.03) × FTIR, in molec/cm$^2$). Such different BIAS for low/high target species concentration levels, due to the presence of both constant and proportional biases, was also recently observed (although with less FTIR sites involved) in another nadir satellite product, the formic acid observed by IASI (Supporting Information in Franco et al. (2020)). The NDACC FTIR network, which covers a large number of atmospheric species at wide ranges of concentrations, is a powerful source of reference

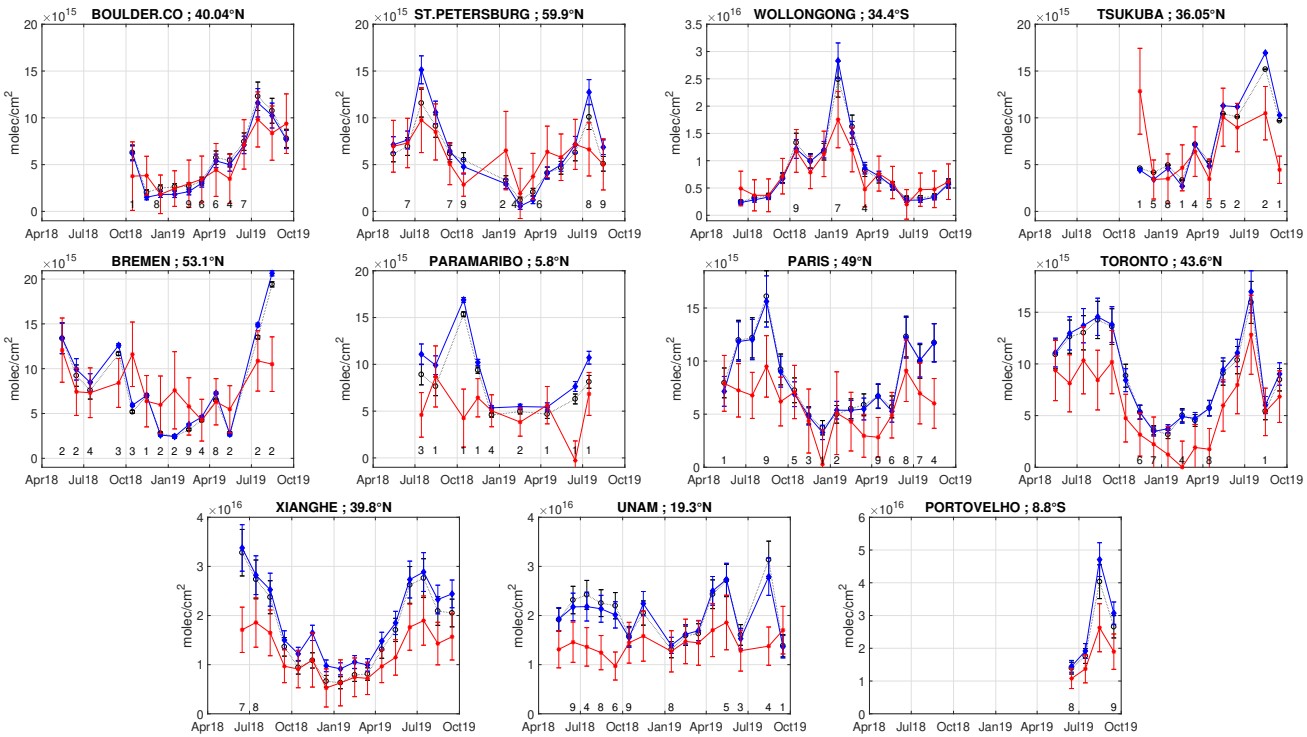

**Figure 5.** *Continued.*

data to detect such nadir satellites' biases.

Although significant, the observed overestimation and underestimation of TROPOMI are within the lower limits of the pre-launch requirements ($\pm 40\%$), as are the biases at individual sites for 20 of the 25 stations. The TROPOMI systematic uncertainty budget is in very good agreement with the observed bias, larger uncertainties being reported at stations where the bias exceeds the 40% requirements. Possible improvements in the TROPOMI biases could be achieved by taking into account aerosol effects over polluted sites, improving the QA values at high SZA, and using an albedo climatology and a priori HCHO profiles at the TROPOMI spatial resolution. Except for the former, these improvements are foreseen in next versions of operational TROPOMI data.

The precision of TROPOMI OFFL products is estimated by the median absolute deviation (MAD) at the clean sites, where the collocation effect is expected to be small. For FTIR HCHO levels lower than $2.5 \times 10^{15}$ molec/cm$^2$, the MAD is $1.3 \times 10^{15}$ molec/cm$^2$, corresponding to a single pixel precision of $7 \times 10^{15}$ molec/cm$^2$ (5 to $8 \times 10^{15}$ at individual sites), which is well below the pre-launch precision requirements of $1.2 \times 10^{16}$ molec/cm$^2$. However, the provided TROPOMI random uncertainties (after launch) were indeed found to be better than the pre-launch requirements, but they are too small by a factor of 1.6 compared to the MAD at the clean sites. There is a factor of 2.3 difference between MAD and the random uncertainty on the comparisons (dominated by TROPOMI random uncertainty) at the high-level sites, where an additional effect of collocation

might take a role as well. The underestimation of the TROPOMI random uncertainty could be due to a random effect of the uncertainty associated to the air mass factor calculation that is not currently included in the budget. This would also explain a larger underestimation of random error at high-levels sites (factor 2.3 vs 1.6 at clean sites). Also, a systematic uncertainty component on a short-term (so not included in the TROPOMI random uncertainty) can have a random effect on our longer-term

comparisons.

We have shown that the TROPOMI data capture very well the HCHO seasonal variability, even at very clean sites. The Pearson correlation coefficient for monthly mean coincident data is 0.91. Although we have found room for a refinement of the TROPOMI random uncertainty estimation and for an improvement of the QA values for a better filtering of the remaining few outliers and negative columns (exceeding the expected statistical distribution), this validation work has demonstrated the

very good quality of the TROPOMI HCHO product, which is well within the pre-launch requirements for both accuracy and precision. This work has also shown the high value of the FTIR HCHO network, providing harmonized and well-characterized data covering a wide range of HCHO columns. These ground-based FTIR data are continuously extended by new measurements and will be used in the coming years for the routine S5P validation within the ESA dedicated validation server (https://mpc-vdaf-server.tropomi.eu/). The FTIR network will also be used in the near future for the validation of previous satellite missions

such as OMI or GOME-2. New FTIR measurements are continuously performed and can be used in the coming years for the validation of new satellite generation, such as TEMPO, GEMS, Sentinel 5P, or Sentinel 4.

An extension of this TROPOMI HCHO validation with ground-based MAX-DOAS and Pandora instruments, especially at sites where both FTIR and UV-Visible techniques are available (e.g. Xianghe, Maïdo, Lauder,...) or at uncovered regions (Africa) would bring additional knowledge. However, there is first a need for a data product harmonization within the MAXD-

OAS network, as was done with the FTIR network used here. This work is ongoing as part of the ESA FRM4DOAS and Pandonia projects.

*Data availability:*

The TROPOMI HCHO data are publicly available at https://scihub.copernicus.eu. The access and use of any Copernicus Sentinel data available through the Copernicus Sentinel Data Hub is governed by the Legal Notice on the use of Copernicus

Sentinel Data and Service Information which is given here:

https://sentinels.copernicus.eu/documents/247904/690755/Sentinel_Data_Legal_Notice.

The FTIR data sets can be provided in the public NDACC repository (ftp://ftp.cpc.ncep.noaa.gov/ndacc/station/, last access: January 2020) depending on each PI decision. Please pay attention to the NDACC data policy. The whole data set used in this publication can be provided upon request to Corinne Vigouroux (corinne.vigouroux@aeronomie.be) and data per station can

be requested from the individual PIs.

*Author contribution:* Corinne Vigouroux and Bavo Langerock performed the validation using HCHO TROPOMI and FTIR data at all sites. They are also involved in the FTIR measurements at Maïdo and Porto Velho. Corinne Vigouroux analyzed the Maïdo, Porto Velho, Sodankylä and Xiangue data. Isabelle De Smedt is the TROPOMI HCHO product lead and participated in the paper (Section 2 and discussions). Zhibin Cheng is the TROPOMI HCHO processor lead. Michel Van Roozendael and

Diego Loyola have a joint responsibility for the TROPOMI HCHO prototype algorithm and operational processor respectively.

Gaia Pinardi was involved in the validation method section through her expertise in validation using UV-Visible techniques, which is part of the projects TROVA and TROVA-2 that funded this work. All other co-authors provided the FTIR HCHO data for the station(s) they are responsible for.

*Competing interests:* The authors declare that they have no conflict of interest.

*Acknowledgements.* This study has been supported by the ESA PRODEX projects TROVA and TROVA-E2 funded by the Belgian Science Policy Office (Belspo). The measurements at Reunion Island have been also supported by the Université de La Réunion and CNRS (LACy-UMR8105 and UMS3365), and at Porto Velho by the BRAIN-pioneer project IKARE, funded by Belspo, with the collaboration of the Instituto Federal de Educaçao, Ciência e Tecnologia de Rondônia (IFRO). SPbU FTIR team has been supported by the Russian Foundation for Basic Research Project #18-05-00011. St. Petersburg FTIR measurements were carried out by the instrumentation of the Geomodel resource

center of SPbU. The NDACC stations Bremen, Izaña, Ny-Ålesund and Paramaribo have been supported by the German Bundesministerium für Wirtschaft und Energie (BMWi) via DLR under grants 50EE1711A, B and D. We thank the EU-project STRATOCLIM for financial support to U. of Bremen. Measurements made at Lauder, by NIWA, are funded by New Zealand's Ministry of Business, Innovation and Employment through the Strategic Science Investment Fund. We thank the AWI Bremerhaven for logistical support and the station personnel in Ny Alesund. We acknowlegde the Meterologische Dienst van Suriname for logistical and on-site support in Paramaribo, Suriname. We

acknowledge the AWI Potsdam and the Coral Reef Foundation for logistical and on-site support in Koror, Palau. ULiège has received support from the F.R.S. - FNRS, from the Fédération Wallonie-Bruxelles and from the GAW-CH programme of MeteoSwiss. We further acknowledge the International Foundation High Altitude Research Stations Jungfraujoch and Gornergrat (HFSJG, Bern) for supporting the facilities needed to perform the Jungfraujoch observations. E. Mahieu is Research Associate with the F.R.S. - FNRS. The National Center for Atmospheric Research is sponsored by the National Science Foundation. The NCAR FTS observation programs at Thule, GR, Boulder, CO,

and Mauna Loa, HI are supported under contract by the National Aeronautics and Space Administration (NASA). The Thule work is also supported by the NSF Office of Polar Programs (OPP). We wish to thank the Danish Meteorological Institute for support at the Thule site and NOAA for support at the Mauna Loa. Financial support from grants by DGAPA-UNAM (07417 & 111418) and CONACYT (290589) are acknowledged as well as the University Network of Atmospheric Observatories (RUOA) for the maintenance and operation of the Mexican stations. The Paris TCCON site has received fundings from Sorbonne Université, the French research center CNRS, the French space agency

CNES and Région Île-de-France. Eureka measurements were made at the Polar Environment Atmospheric Research Laboratory (PEARL) under the CANDAC and PAHA projects led by James R. Drummond, and in part by the Canadian Arctic ACE/OSIRIS Validation Campaigns, led by Kaley A. Walker. Funding was provided by AIF/NSRIT, CFI, CFCAS, CSA, ECCC, GOC-IPY, NSERC, NSTP, OIT, PCSP, and ORF. Logistical and operational support was provided by PEARL Site Manager Pierre Fogal, the CANDAC operators, and the ECCC Weather Station. Toronto measurements were made at the University of Toronto Atmospheric Observatory, supported by CFCAS, ABB Bomem, CFI,

CSA, ECCC, NSERC, ORDCF, PREA, and the University of Toronto. FTIR operations of the Rikubetsu and Tsukuba sites are financially supported in part by the GOSAT series project.

    The authors would like to thank all the people responsible for the FTIR measurements and/or data analysis at the different sites: C. Hermans, N. Kumps, M. Zhou, BIRA-IASB; L. Gatti, INPE; Uwe Raffalski, IRF Kiruna; Omaira García and Eliezer Sepulveda, AEMet; C. Becker, Paramaribo; J. Robinson, Lauder; A.V. Poberovskii, H.H. Imkhasin, S.I. Osipov, SPbU; A. Bezanilla and C. Guarín, UNAM; P.

Jeseck, Sorbonne Université; M. Rettinger, IMK-IFU; H. Nakajima, NIES; C. Servais: Université de Liège.

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
