# Peer review of "TROPOMI/S5P formaldehyde validation using an extensive network of ground-based FTIR stations"

_Atmospheric Measurement Techniques, 2020_

## Referee Comment (RC1) · Anonymous Referee #1 · 17 Mar 2020

**Review: TROPOMI/S5P formaldehyde validation using an extensive network of ground-based FTIR stations by Vigouroux et al.**

**General comments**

Vigouroux and co-authors present extensive validation of TROPOMI's formaldehyde retrievals (version 1.1.[5-7]) against ground-based FTIR retrievals from 25 stations around the world. Most of this stations belong to the Network for the Detection of Atmospheric Composition Chance (NDACC). They results indicate that TROPOMI satisfies pre-launch precision and accuracy requirements. TROPOMI overestimates HCHO columns (~26%) over locations with small HCHO levels while underestimates HCHO columns (~30%) over locations with high HCHO levels.

This paper provides an excellent example of careful and sound satellite validation using ground-based remote sensing observations. Provides a detailed description of the methods and datasets used. It is well written and provides clear descriptions of the most important results. The paper should be publish with minimal changes since it provides a compelling case supporting the quality and capacity of S5P HCHO retrievals, its current biases and what users should expect to achieve with S5P.

Some minor questions are raised. The aim is to further improve the clarity of the text and the description of the methodology and results.

**Specific comments**

Abstract.

> Page 2, line 4: "We observe that, at all sites, the TROPOMI accuracy is below the upper limit of the pre-launch requirements of 80%, and below the lower limit of 40% for 20 of the 25 stations." This sentence is confusing. What are the pre-launch requirements? If HCHO retrievals accuracy are below lower limit of 40% there are also below the upper limit of 80% why both are mentioned?

Introduction.

> Page 3, line 2: Validation from aircraft has been expanded to multiple locations by Zhu et al., 2020 (https://www.atmos-chem-phys-discuss.net/acp-2019-1117/). Could we valuable to add it to the list of aircraft based validation efforts?

TROPOMI HCHO data.

> The description of TROPOMI data and versions is very complete but after reading this section the question remains, off all the options (RPRO, OFFL and NRTI) which one has

been used? If several depending on the station and the period of time, that should also be explained?

Given the unprecedented TROPOMI spatial resolution, the surface elevation could play a bigger role while explaining biases for some locations with complicated topography. What is the source of TROPOMI surface elevation information?

Page 4, line 6. "All cross-sections have been pre-convolved", these cross-section include HCHO and interferers but that may be not clear to someone without a background on DOAS retrievals. Maybe worth explaining? How stable have been TROPOMI slit functions after lunch? Is the algorithm correcting cross-sections for changes in the slit function?

Page 4, line 20. How is $M_0$ calculated? Is it the average of the AMFs of the slant columns considered in the calculation of $N_{(s,0)}$?

Ground-based FTIR HCHO data

Figure 1 caption could be expanded to provide some information about the spatial resolution of the averaged TROPOMI data shown. What kind of averaging algorithm was used to generate the background data?

Page 7, line 22:  Maybe adding described by to "is 13% in the network of Vigouroux et al., (2018)" could be more precise "is 13% in the network described by Vigouroux et al., (2018)"

Page 7, line 25: Please clarify, it looks like if stations using the PROFFIT9 retrieval code can have bigger systematic uncertainty due to uncertainty on the channeling that is not taken into account yet in the SFIT4 code. If the SFIT4 code is not taking this channeling uncertainty in the budget it just means that is introducing a systematic error for those stations?

Page 8, line 3: Why the smoothing systematic uncertainty (on the total column) is significantly bigger for the 5 added sites?

Validation method

Collocation criteria

What is the effect of reducing/increasing the TROPOMI/FTIR collocation radius (currently set at 20km)? Is there a radius threshold/range where no improvement is achieved in the comparisons?

For each station, after co-adding, what is the median TROPOMI detection limit and random uncertainty? That will be an interesting fact to know

Building inter-comparable products

Equation 2 could have dimensions problem: $\mathbf{a}_S$ SP5 averaging Kernel is defined on the S5P vertical grid according to line 16 page 9 while $x'_F$ and $x_{S,a}$ are defined on the FTIR vertical grid.

Validation results

As mentioned above, including a table showing the period of time each one of the products (RPRO, OFFL) has been used in the calculations will assure full reproducibility of the results shown.

TROPOMI observed BIAS and accuracy

Page 12, line 10: This sentence is confusing "… it is negative for higher levels and very consistent for the stations from 8.7 to 28.6 x $10^{15}$…" This is my interpretation "… it is negative and very consistent for stations with higher levels, ranging from 8.7 to 28.6 x $10^{15}$…" but maybe is the HCHO level what is 8.7 to 28.6 x $10^{15}$.

Page 12, line 10: Lower levels are defined in the abstract and below at page 12, line 21 as 2.5x$10^{15}$ molec/cm$^2$. What is the meaning of 6.5x$10^{15}$ molec/cm$^2$.

Do the authors suggestions on how to link/explain the constant and proportional biases to different instrumental, algorithm, or geophysical parameters?

---

## Referee Comment (RC2) · Anonymous Referee #2 · 30 Mar 2020

**TROPOMI/S5P formaldehyde validation using an extensive network of ground-based FTIR stations**

Corinne Vigouroux , Bavo Langerock, Carlos Augusto Bauer Aquino, Thomas Blumenstock, Martine De Mazière, Isabelle De Smedt, Michel Grutter, James Hannigan, Nicholas Jones, Rigel Kivi, Erik Lutsch8, Emmanuel Mahieu, Maria Makarova, Jean-Marc Metzger, Isamu Morino, Isao Murata, Tomoo Nagahama, Justus Notholt, Ivan Ortega, Mathias Palm, Gaia Pinardi, Amelie Röhling, Dan Smale, Wolfgang Stremme, Kim Strong, Ralf Sussmann, Yao Té, Michel van Roozendael, Pucai Wang, and Holger Winkler

This paper describe an extensive validation of HCHO TROPOMI operational product (version 1.1.[5-7]) by using twenty five ground-based solar absorption FTIR stations around the world, most of them affiliated to NDACC. The results found an overestimation of +26±5% of TROPOMI products for columns below 2.5x10$^{15}$ molec/cm$^2$ and an underestimation of about -30.8±1.4% for columns larger than 8x10$^{15}$ molec/cm$^2$. These results satisfy the pre-lunch requirements for TROPOMI. They present clear and detail description of the method used for the validation between satellite and ground-based measurements.

Although, the main finding are very well described, my main concern with the paper is the missing discussion on the reasons of main difference between TROPOMI and FTIR formaldehyde BIAS for some stations (large offsets) and also difference in seasonal cycle (e.g. Paramaribo, Paris, UNAM…) (See Figure 5.).

The topic of this work fits well within the scope of AMT. Although the paper is well structured, the text needs to be carefully revised in order to be more precise in some sections.

I recommend acceptance to AMT after addressing the comments above and few minor comments below.

Page 2, line 5, confusing sentence, "accuracy is below the upper limit of the pre-launch requirements of 80%, and below the lower limit of 40% for 20 of the 25 stations", it does not make sense to write that HCHO TROPOMI retrievals are below lower and upper limits. Please clarify it.

Page 3, line 1, is there any study of validation of satellite HCHO observation with ship-based measurements?

Page 3, line 8, please define what is "TROPOMI Cal/Val"

.Page 3, line 15, would you please mention what are the differences among versions from v.1.1.5 to v.1.1.7?

Page 4, line 12, why to use OMI albedo climatology?

Page 4, line 13, "(Kleippol et al., 2008)".

Page 4, line 20, please define all the quantities of the equation (e.g., M and M0)

Page 6, line 6, what is the main difference between PROFITT9 and SFIT4.0.9.4?

Page 7, line 7, please be consistent between names used in the text "Maïdo" and used in the figure 1.

Page 8, line 3,what are the reasons for the lowest smoothing systematic uncertainties in the 5 added sites.

Page 8, line 25, please remove "so"

Page 11, line 29, would be nice if you include one or two sentences describing the main differences between OFFL, RPRO and NRTI products. Are they different at all.

Page 16, line 31, would you please clarify how the collocation plays a role in Maïdo? Fire emissions are included in the calculation of the a-priori profiles? Could fire emissions enhanced the HCHO amounts? What is the effect of changing the collocation radius in this station?

---

## Author Comment (AC1) · 4 May 2020

**Reply to Anonymous Referee #1**

**General comments**

**Vigouroux and co-authors present extensive validation of TROPOMI's formaldehyde retrievals (version 1.1.[5-7]) against ground-based FTIR retrievals from 25 stations around the world. Most of this stations belong to the Network for the Detection of Atmospheric Composition Chance (NDACC).They results indicate that TROPOMI satisfies pre-launch precision and accuracy requirements. TROPOMI overestimates HCHO columns (~26%) over locations with small HCHO levels while underestimates HCHO columns (~30%) over locations with high HCHO levels.**

**This paper provides an excellent example of careful and sound satellite validation using ground-based remote sensing observations. Provides a detailed description of the methods and datasets used. It is well written and provides clear descriptions of the most important results. The paper should be publish with minimal changes since it provides a compelling case supporting the quality and capacity of S5P HCHO retrievals, its current biases and what users should expect to achieve with S5P.**

**Some minor questions are raised. The aim is to further improve the clarity of the text and the description of the methodology and results.**

We thank the referee for their very positive review and for their work that is helping us to improve the manuscript.

**Specific comments**

**Abstract.**

**Page 2, line 4:"We observe that, at all sites, the TROPOMI accuracy is below the upper limit of the pre-launch requirements of 80%, and below the lower limit of 40% for 20 of the 25 stations." This sentence is confusing. What are the pre-launch requirements? If HCHO retrievals accuracy are below lower limit of 40% there are also below the upper limit of 80% why both are mentioned?**

The TROPOMI accuracy pre-launch requirements are given as a range: "40-80%". We have distinguished between the two limits of the ranges because at all sites the 80% requirements are reached (but this is an upper limit for the expected TROPOMI accuracy), and at 20 of the 25 sites the lower limit of the range (40%) is reached. Therefore, at 5 sites, we have a bias between 40 and 80%. To avoid any confusion, we have rewritten it as follows:

*"The pre-launch requirements of the TROPOMI HCHO accuracy are 40-80%. We observe that these requirements are well reached, with the BIAS found below 80% at all the sites, and below 40% at 20 of the 25 sites."*

**Introduction.**

**Page 3, line 2: Validation from aircraft has been expanded to multiple locations by Zhu et al., 2020 (https://www.atmos-chem-phys-discuss.net/acp-2019-1117/). Could we valuable to add it to the list of aircraft based validation efforts?**

Indeed. This reference has been added in the manuscript.

**TROPOMI HCHO data.**

**The description of TROPOMI data and versions is very complete but after reading this section the question remains, off all the options (RPRO, OFFL and NRTI) which one has been used? If several depending on the station and the period of time, that should also be explained?**

The text in our AMTD version was:
"At the time of writing this paper, the latest product versions 1.1.[5-7] provide a consistent time series of Reprocessed+Offline (RPRO+OFFL) data, covering the period between May 2018 up to (at least) December 2019 (last access). The Near-Real-Time (NRTI) product, for the same versions 1.1.[5-7], started in December 2018. Details are found in the Readme file (http://www.tropomi.eu/sites/default/files/files/publicSentinel-5P-Formaldehyde-Readme.pdf; doi: 10.5270/S5P-tjlxfd2)."

Indeed, the referee is right: it is not clear in this TROPOMI section which products are used in this paper (RPRO + OFFL, or NRTI). Actually, we performed the validation on the two sets of data. But, in this paper the tables and figures focus on the RPRO+OFFL data set. The NRTI validation results are so similar that we preferred avoiding giving details on them. We only give a summary of the NRTI biases in Sect. 5.1.

At all sites, the TROPOMI data set that we used is a combination of RPRO and OFFL products, from v.1.1.5 to 1.1.7, the versions 5 to 7 being consistent retrieved HCHO products. Indeed, the number of version corresponds to different period of time, but we did not find relevant to detail them since the products are consistent among these versions. The details of the dates are in the Readme file (more precisely in its Table 2) for which we gave the reference. For the referee and readers convenience, we provide them here, and *we will repeat them in a Table in the next version of the manuscript:*

- *From 2018-05-14 to 2018-11-28 : RPRO v.1.1.5*
- *From 2018-11-28 to 2019-03-28 : OFFL v.1.1.5*
- *From 2019-03-28 to 2019-04-23 : OFFL v.1.1.6*
- *From 2019-04-23 to present      : OFFL v.1.1.7*

    The validation of the NRTI products (results only summarized in one sentence in Sect. 5.1) is using:

- From 2018-12-05 to 2019-04-04 : NRTI v.1.1.5
- From 2019-04-04 to 2019-04-30 : NRTI v.1.1.6
- From 2019-04-30 to present      : NRTI v.1.1.7

We have also repeated in the new table (on request of referee#2), the information on the differences between the versions that is in the ReadMe file.

We have added to the manuscript (in italic):
"At the time of writing this paper, the latest product versions 1.1.[5-7] provide a consistent time series of Reprocessed+Offline (RPRO+OFFL) data, covering the period between May 2018 up to (at least) December 2019 (last access). *The detailed validation results shown in Sect. 5 are obtained using this consistent time-series (RPRO+OFFL, from 2018-05-14 to 2019-12-31). The version numbers and their dates of change are given in Table 1, and further details are given in the Readme file* (http://www.tropomi.eu/sites/default/files/files/publicSentinel-5P-Formaldehyde-Readme.pdf; doi: 10.5270/S5P-tjlxfd2). The Near-Real-Time (NRTI) product, for the same versions 1.1.[5-7], started in December 2018 *up to December 2019 (last access). This product has also been validated, but the results being very similar to the RPRO+OFFL validation, we do not show them in details in this paper."*

**Given the unprecedented TROPOMI spatial resolution, the surface elevation could play a bigger role while explaining biases for some locations with complicated topography. What is the source of TROPOMI surface elevation information?**

Yes, we agree that topography could play a significant role if not taken into account carefully, both for the quality of the product, and for the comparison between satellite and ground-based quantities. However, we considered it in both cases.
For S5P L2 products, the digital elevation map is from GMTED2010 (Danielson et al., 2011), and an average over the ground pixel area is considered. Furthermore, as explained in the HCHO the Algorithm Theoretical Basis Document (ATBD, De Smedt et al. 2018): "To reduce the errors associated to topography and the lower spatial resolution of the model compared to the TROPOMI 3.5x7 km2 spatial resolution, the a priori profiles need to be rescaled to effective surface elevation of the satellite pixel. The TM5 surface pressure is converted by applying the hypsometric equation and the assumption that temperature changes linearly with height"
Finally, as described in Sect.4.2, the different elevation between the altitude of the ground-based station and the surface elevation of the satellite pixel is taken into account. We believe that the positive bias usually observed at mountain stations is related to the constant bias of TROPOMI for small HCHO columns, because it is also observed at clean sites that have an altitude close to sea level (Kiruna, Ny-Alesund).

Danielson, J.J., and Gesch, D.B.: Global multi-resolution terrain elevation data 2010 (GMTED2010): U.S. Geological Survey Open-File Report 2011–1073, 26 p, 2011.

De Smedt, I., Theys, N., Yu, H., Danckaert, T., Lerot, C., Compernolle, S., Van Roozendael, M., Richter, A., Hilboll, A., Peters, E., Pedergnana, M., Loyola, D., Beirle, S., Wagner, T., Eskes, H., van Geffen, J., Boersma, K. F., and Veefkind, P.: Algorithm theoret10 ical baseline for formaldehyde retrievals from S5P TROPOMI and from the QA4ECV project, Atmos. Meas. Tech., 11, 2395–2426, https://doi.org/10.5194/amt-11-2395-2018, 2018.

**Page 4, line 6. "All cross-sections have been pre-convolved", these cross-sections include HCHO and interferers but that may be not clear to someone without a background on DOAS retrievals. Maybe worth explaining? How stable have been TROPOMI slit functions after launch? Is the algorithm correcting cross-sections for changes in the slit function?**

Together with the HCHO cross-section, the absorptions of NO2, BrO, O3 (at two temperatures) and O4 are fitted. A Ring cross-section and two pseudo-cross sections to account for non-linear O3 absorption effects are also included in the fit. References are given in De Smedt et al. (2018). *This more detailed description has been added in the new manuscript.*

The operational algorithm does not have the capability to fit directly the slit functions, it has to be done offline. Up to now, the TROPOMI slit functions have been stable. No update of the pre-convolved cross-sections are planned, but this is monitored.

**Page 4, line 20. How is M0 calculated? Is it the average of the AMFs of the slant columns considered in the calculation of N(s,0)?**

Yes; M0 is an average of the air mass factors (M) of the slant columns selected in the reference sector, the Pacific Ocean (N(s,0)).
*This has been added in the text.*

**Ground-based FTIR HCHO data**

**Figure 1 caption could be expanded to provide some information about the spatial resolution of the averaged TROPOMI data shown. What kind of averaging algorithm was used to generate the background data?**

The spatial resolution used for this map is 0.2°x0.2°. We use the HARP v1.5 tool, which can be found at https://atmospherictoolbox.org. This information has been added in the Fig.1 caption, as suggested by the referee.

**Page 7, line 22: Maybe adding described by to "is 13% in the network of Vigouroux et al., (2018)"could be more precise "is 13% in the network described by Vigouroux et al., (2018)"**

Done, as suggested.

**Page 7, line 25: Please clarify, it looks like if stations using the PROFFIT9 retrieval code can have bigger systematic uncertainty due to uncertainty on the channeling that is not taken into account yet in the SFIT4 code. If the SFIT4 code is not taking this channeling uncertainty in the budget it just means that is introducing a systematic error for those stations?**

The channeling is due to (possible) imperfections in the instrument that may (or may not) lead to artefacts in the interferogram. This error is included in the PROFFIT9 code, and not yet in SFIT4. However, at present the fact that there is or not a channeling in the spectra at each station (it is obviously depending on each instrument) has not been measured at each site. Such an exercise has been initiated after the Vigouroux et al. (2018) paper for a set of stations (by T. Blumenstock, KIT, co-author of the present paper), but has not been done at each site systematically. For the sites that have been tested, we found that a non-negligible channeling is indeed present at some sites, but not at all sites. Therefore, introducing such an additional error in the theoretical calculation without knowing if it is indeed present may also lead to an overestimation of the

systematic uncertainty. In the next update version of SFIT4, the random and systematic error on the target species due to channeling will be included, but its correct estimation would be possible only at the sites where the channeling itself is estimated. This is an on-going work within the IRWG (InfraRed Working Group) of NDACC.

In the present validation, the systematic bias between TROPOMI and FTIR stations are very consistent among the stations (see Fig. 3), except for Eureka which is the only clean site with a negative TROPOMI BIAS. However, Eureka was one of the sites participating on the channeling exercise, and the channeling was found very small for this instrument. So the channeling error is not explaining the different bias there. For the other stations, the good consistency of the TROPOMI BIAS at the different stations (which depends on the HCHO levels, and not on individual sites), shows that the BIAS is dominated by the TROPOMI systematic error, and that the channeling one should have a smaller impact.

To clarify that the channeling is not always under-estimated in the SFIT4 stations, and can be over-estimated in some PROFFIT4 stations, we have adapted the text:
*"The systematic uncertainty can be larger (up to 21-26%) at the stations using the PROFFIT9 retrieval code, due to an assumed uncertainty on the channeling that is not taken into account yet in the SFIT4 code. However, this channeling uncertainty can also be negligible at some sites (it depends on each instrument), and more investigation is needed at each station to avoid its under- or over-estimation."*

**Page 8, line 3: Why the smoothing systematic uncertainty (on the total column) is significantly bigger for the 5 added sites?**

We think the referee has misinterpreted the sentence. The 13% and 14% for the 5 added sites, are for the total systematic uncertainty (dominated by the spectroscopy), and not for the smoothing part only. To avoid the confusion, we have changed the sentence to :
*"For the five added sites, the median total systematic uncertainty is 13% (Jungfraujoch, Tsukuba, Palau), or 14% (Rikubetsu, Xianghe), commensurate with the other sites."*

**Validation method**

**Collocation criteria**

**What is the effect of reducing/increasing the TROPOMI/FTIR collocation radius (currently set at 20km)? Is there a radius threshold/range where no improvement is achieved in the comparisons?**

Before choosing the 20km collocation radius, we have indeed tested several distances: 10, 20, 30, 40, and 50 km. We provide in this discussion a plot of the median relative differences (bias) at each station (Fig.1) for the different collocation distances. Please, note that the numbers are not the same as in the AMTD paper, because this work on collocation distances were made in the course of the project (not at the time of writing the paper), so the time-series were shorter, and the collocated time was 6h (now it is set to 3h). We see in Fig. 1, usually similar biases for the 20 to 50 km criteria, especially for mid- HCHO levels sites. For clean sites, we observe usually slightly smaller biases with the 30km criteria than with the 20km one. For the most polluted sites, UNAM (Mexico City) and Porto Velho, the bias is increasing with the distance. The 10km collocation leads to more than twice less coincidences (at some stations, even 5 times less).

Therefore, the median biases obtained with this criterion were less robust, and the 10km choice was discarded.

[Figure]

Figure 1: Median bias at each station for the different collocation distances. The numbers in black are the number of coincidences, from the 10km criterion (top) to the 50km criterion (bottom).

The median biases, being usually similar using the different collocation distances, were not so useful to determine our choice of collocation. We therefore looked at the MAD (median absolute deviation, see Eq. 6 for complete definition) to help for the choice. Figure 2 shows the MAD at each station for the different collocation distance.

[Figure]

Figure 2: MAD at each station for the different collocation distances. The numbers in black are the number of coincidences, from the 10km criterion (top) to the 50km criterion (bottom).

We see from the figure that usually the MAD is decreasing with the distance increasing, except at a few cases (the polluted cases as expected: Porto Velho, UNAM=Mexico City,…). However, we cannot conclude that the comparisons are "improved": indeed, while the MAD is decreasing due to the averaging of more TROPOMI pixels, the random uncertainties of the comparisons are also decreasing. In a world where the random error would be perfectly determined, we would have a constant ratio MAD / RandErr (no dependence on the collocation distance), equal to 1 if

there is no collocation effect (so expected to be 1 at clean sites). If we plot this ratio (Figure 3), we see that it is increasing with the distance, pointing to an additional random error due to the collocation.

[Figure]

Figure 3: The ratio between the MAD and the random uncertainty on the differences between TROPOMI and FTIR.

As no clear threshold provides an improvement of the comparisons, we therefore decided to use the 20km collocation choice, a good compromise between the number of coincidences, and the best correspondence between MAD and random uncertainty budget. It also avoids an increasing MAD over the highest HCHO level sites (UNAM, Porto Velho).

In the new manuscript, we summarize this study by adding the following text:
"Before choosing the 20 km collocation criteria, we have tested several distances (10, 20, 30, 40, and 50 km). The 10 km criterion was discarded because of the poor remaining coincidences leading to less robust statistics. The 20 to 50 km criteria give similar biases between TROPOMI and FTIR. The standard deviations of the comparisons usually decrease slightly with increasing collocation distance due to a smaller TROPOMI random uncertainty (more pixels to average), except at the most polluted sites. However, the ratio between the standard deviations and the random uncertainty budgets is increasing with the collocation distance at all sites, pointing to an increased random error due to the collocation. We therefore choose the 20~km distance to reduce the random spatial collocation error."

**For each station, after co-adding, what is the median TROPOMI detection limit and random uncertainty? That will be an interesting fact to know**

In Table 2 of the AMTD paper, we give $\sigma_{rand}$ for each station. This value is the random uncertainty on the differences between TROPOMI and FTIR. It is fully defined by Eq. 7. In the text (Sect 4.3), we explain that since the other terms of Eq.7 are much smaller, $\sigma_{rand}$ is dominated by the TROPOMI random error budget $\sigma_{S,rand}$. Therefore, the $\sigma_{rand}$ in Table 2 is in first approximation the number that the referee is asking (~TROPOMI random uncertainty, $\sigma_{S,rand}$). Please note that there was an error in AMTD version in Eq. 7: the matrix for FTIR random uncertainty was called $S_{s,rand}$ instead of $S_{F,rand}$. It is now corrected.

Then, the detection limit is usually defined as $3*\sigma_{S,rand}$, so it is easily determined at each station from Table 2, by approximating $\sigma_{S,rand}$ with the provided $\sigma_{rand}$, and multiplying by 3. For all stations together, we obtain $3.6 \times 10^{15}$ molec/cm$^2$ as the TROPOMI detection limit (for an average of about 34 pixels), so $2.1 \times 10^{16}$ molec/cm$^2$ for a single pixel.

**Building inter-comparable products**

**Equation 2 could have dimensions problem: $a_S$ SP5 averaging Kernel is defined on the S5P vertical grid according to line 16 page 9 while $x'_F$ and $x_{S,a}$ are defined on the FTIR vertical grid.**

Actually, we said in the text above Eq. 2, that $x'_F$ has been regridded to the satellite grid before applying Eq. 2. But the referee is right that this is not clear enough because we kept the same name for $x'_F$ and $x_{S,a}$ in both grids (to try to have a small number of variable names). So, for clarity, we now introduce different names for different grids: we call now $x_{S,a}$ the S5P a priori on the original satellite grid and keep $x'_F$ the FTIR profile on original FTIR grid, and we call $x_{S,a/F}$ the S5P a priori profile regridded to the FTIR grid, and $x'_{F/S}$ the FTIR profile regridded to the satellite grid.

The new text becomes:

"*First, the a priori substitution is applied, using the S5P a priori profile as the common a priori profile. For this, the S5P a priori profile xS,a is regridded to the FTIR retrieval grid (xS,a/F) using a mass conservation algorithm (Langerock et al., 2015). In the rare situation where the satellite pixel elevation is above the FTIR site, the S5P a priori profile is extended to the FTIR instrument's altitude. The regridded S5P a priori xS,a/F is then substituted following Rodgers and Connor (2003), and we finally use the corrected FTIR retrieved profile x'F in the comparisons:*

$x'_F = x_F + (A_F - I)(x_{F,a} - \underline{x_{S,a/F}})$,

where …"

And also below:

"*For that purpose we regrid the corrected FTIR profile x'F to the S5P column averaging kernel grid (x'F/S) and apply the smoothing equation:*

$c_F^{smoo} = c_{S,a} + a_S(\underline{x'_{F/S}} - \underline{x_{S,a}})$       (2)

*with $c_{S,a}$ the S5P a priori column derived from the S5P a priori profile. We obtain a smoothed FTIR column $c_F^{smoo}$ associated with a collocated TROPOMI pixel. In the case of mountain sites where the pixel altitude is below the instrument's height, the regridding of the FTIR profile x'F/S is done…*"

**Validation results**

**As mentioned above, including a table showing the period of time each one of the products (RPRO, OFFL) has been used in the calculations will assure full reproducibility of the results shown.**

We followed the referee's suggestion by adding such a Table (now Table 1).

**TROPOMI observed BIAS and accuracy**

**Page 12, line 10: This sentence is confusing "...it is negative for higher levels and very consistent for the stations from 8.7 to 28.6 x 10$^{15}$..."This is my interpretation "...it is negative and very consistent for stations with higher levels, ranging from 8.7 to 28.6 x 10$^{15}$..." but maybe is the HCHO level what is 8.7 to 28.6 x 10$^{15}$.**

**Page 12, line 10: Lower levels are defined in the abstract and below at page 12, line 21 as 2.5x10$^{15}$ molec/cm$^2$. What is the meaning of 6.5x10$^{15}$ molec/cm$^2$.**

We meant that the biases were always negative above 6.5x10$^{15}$ (including Tsukuba and Bremen), and that they are consistent "only" above 8.7 x10$^{15}$ (because the bias at Bremen, -5%, is lower). The 6.5x10$^{15}$ limit was appearing in Table 2 (AMTD version) as a limit between positive/non significant trends (below) and always negative trends (above). However, because the limit of 8.0 x10$^{15}$ was chosen for the "high levels" median bias calculation, we did not put a separation line at 6.5x10$^{15}$, which seems to be source of confusion. We decided to simplify the sentence as suggested by the referee.

**Do the authors suggestions on how to link/explain the constant and proportional biases to different instrumental, algorithm, or geophysical parameters**

This validation exercise could not identify a specific problem in the instrument itself or in the satellite retrieval algorithm. We will add the following text to the new manuscript (end of Sect. 5.1) in order to give some possible explanations to the observed biases (that are, however, in agreement with the systematic uncertainty budget).

The systematic uncertainties leading to the observed constant and proportional biases of our study have been calculated as described in Sect. 3 of De Smedt et al. (2018). From the error propagation of the HCHO TROPOMI columns (equation of $N_v$, in Sect. 2 of our AMTD paper, now numbered Eq.1 in the new manuscript), it can be found that the proportional bias is more likely due to air mass factor ($M$) uncertainties $\sigma_M$, while the constant bias is more likely due to the uncertainties of the slant columns uncertainties $\sigma_{N,S}$ and to the uncertainty of the background correction of the slant columns. This can be seen in Eq. 13 of De Smedt et al. (2018), where $\sigma_M$ is proportional to $N_s-N_{s,0}$.

We can list some known difficulties of the satellite product:

- The negative bias over high HCHO levels sites (biomass burning or megacities) could be due to aerosol effects. There is no plan to include a correction for aerosols in the operational product, but specific studies are foreseen to check its impact in a scientific product.

- The positive bias over clean polar sites could be due to the solar zenith angle (SZA) dependency of the slant columns fit results (because of spectral interferences with ozone

and BrO). As explained in the paper, the QA values need to be improved at large SZA, which is foreseen in the next version.

- The current albedo climatology is too coarse for TROPOMI, which could be especially a problem for polar, mountain or coastal sites. A climatology based on TROPOMI measurements is under development.
- It is also foreseen to test a regional model at higher spatial resolution for an improvement of the a priori HCHO profiles. This should improve the TROPOMI retrieved product, especially at polluted sites. However, the validation presented here is already taking the a priori information and averaging kernels into account. We therefore do not expect an important effect of the improved a priori profiles on the validation results.

In the conclusion, we have added the following summary:

Possible improvements in the TROPOMI biases could be achieved by taking into account aerosol effects over polluted sites, improving the QA values at high SZA, and using an albedo climatology and a priori HCHO profiles at the TROPOMI spatial resolution. Except for the former, these improvements are foreseen in next versions of the operational TROPOMI data.

---

## Author Comment (AC2) · 4 May 2020

**Reply to Anonymous Referee #2**

**General comments**

**Although, the main finding are very well described, my main concern with the paper is the missing discussion on the reasons of main difference between TROPOMI and FTIR formaldehyde BIAS for some stations (large offsets) and also difference in seasonal cycle (e.g. Paramaribo, Paris, UNAM…) (See Figure 5.).**

**The topic of this work fits well within the scope of AMT. Although the paper is well structured, the text needs to be carefully revised in order to be more precise in some sections. I recommend acceptance to AMT after addressing the comments above and few minor comments below.**

We thank the referee for their work and useful comments.

We first answer on the main remarks above and reply then following the minor comments below.

We have added some possible reasons for the observed TROPOMI bias in the revised version of the manuscript. To avoid repetition, we refer to our reply to referee#1 (last page) who had the same concern as referee#2 on missing discussion on the observed biases.
It should be noted that even if the offsets are large, they are within the accuracy requirements of the satellite (which were based on previous validation studies of HCHO satellite measurements), meaning that such large biases were expected.

The TROPOMI and FTIR seasonal cycles are usually in agreement. However, as pointed out by the referee, this is not the case for Paramaribo. But, as can be seen in Fig.5, the sampling (number of coincidences) is bad there with often only one coincidence per month. Then, if TROPOMI has a remaining outlier, it has a strong influence on the plotted seasonal cycle (e.g. June 2019 shows a negative TROPOMI value). With an improved QA value as expected for the next TROPOMI versions, the comparisons should also improve. For Paris, UNAM (Mexico City) and usually all polluted sites, the TROPOMI and FTIR seasonal cycles show similar features, but the amplitude is smaller with TROPOMI due to its proportional bias that leads to more under-estimation for high HCHO levels (so more under-estimation during the maximum of the FTIR seasonal cycle).

**Page 2, line 5, confusing sentence, "accuracy is below the upper limit of the pre-launch requirements of 80%, and below the lower limit of 40% for 20 of the 25 stations", it does not make sense to write that HCHO TROPOMI retrievals are below lower and upper limits. Please clarify it.**

We have clarified the text:
*"The pre-launch requirements of the TROPOMI HCHO accuracy are 40-80%. We observe that these requirements are well reached, with the BIAS found below 80% at all the sites, and below 40% at 20 of the 25 sites."*

**Page 3, line 1, is there any study of validation of satellite HCHO observation with ship-based measurements?**

Indeed. We have added two references as example of such studies (Peters et al., 2012; Tan et al., 2018).

**Page 3, line 8, please define what is "TROPOMI Cal/Val"**

Done.

**Page 3, line 15, would you please mention what are the differences among versions from v.1.1.5 to v.1.1.7?**

In the AMTD paper, we referred to the ReadMe file (https://sentinel.esa.int/documents/247904/3541451/Sentinel-5P-Formaldehyde-Readme.pdf) for details on the differences in the versions because they have minor impacts on the HCHO TROPOMI time-series. However, as both referees ask that all is included in our paper, we have included a Table (Table 1 in the updated version) repeating the information about the different versions (dates and changes). (see also reply to referee#1)

**Page 4, line 12, why to use OMI albedo climatology?**

The OMI albedo climatology is the best product existing at 340 nm. The spatial resolution is indeed too coarse for TROPOMI. We are waiting for a climatology based directly on TROPOMI, but it is not yet available.

**Page 4, line 13, "(Kleippol et al., 2008)".**

Done (changed to Kleipool).

**Page 4, line 20, please define all the quantities of the equation (e.g., M and M0)**

All quantities have been defined in the text above the equation, except M0, which is an average of the air mass factors (M) of the slant columns selected in the reference sector, the Pacific Ocean (N(s,0)). We have added its definition in the new manuscript.

**Page 6, line 6, what is the main difference between PROFITT9 and SFIT4.0.9.4?**

Both codes are very similar. They are both line-by-line models for infrared solar transmittance spectra, including a radiative transfer model (FSCATM and KOPRA for SFIT4 and PROFITT9, respectively), and based on the optimal estimation method (Rodgers, 2000). They both allow for the Tikhonov regularization as well. Differences are minor, mainly lying in the different options that are available (but not used in the present work) in PROFFIT9: e.g. possibility to retrieve the temperature profiles,… The only relevant difference for the present work is the calculation of channeling error that is not yet included in SFIT4.
The use of different codes within the InfraRed Working Group (IRWG) of NDACC is historical. To certify a good homogenization in the delivered FTIR products in the NDACC database,

harmonization in the retrieved parameters (spectral micro-windows, a priori profiles, spectroscopic database,…) is required for all NDACC target species, and has also been done for the HCHO products presented here (Vigouroux et al., AMT, 2018). A comparison exercise of the two codes has been performed for four species (Hase, JQSRT, 2004) and an agreement within 1% has been found in the retrieved columns. Further details on both codes are available in this latter reference, which has been added in the next version of our manuscript.

**Page 7, line 7, please be consistent between names used in the text "Maïdo" and used in the figure 1.**

The figures are automatically generated using the name provided by the PIs in the geoms file. Indeed, it might confuse the reader to see two names, maybe not so much for Maïdo / LA.REUNION.MAIDO; but for Mexico City (used in Tables and text) and UNAM (used in automated figures). To help the reader, we have explicitly added the two possible names in Table 1 (Table 2 in the new version). We prefer to keep using both names because a geoms data user would find the "UNAM" name for the station, while a "Mexico City" name makes the information clearer for a simple reader that the station is in the city center of Mexico. Note that this situation is also there for other stations, but with clear signification (Izaña / IZANA; Mauna Loa / MAUNA.LOA.HI;…).
Because, the correspondence is less clear for Mexico City / UNAM, we have also repeated the two names in Table 2 (Table 3 in the new version).

**Page 8, line 3,what are the reasons for the lowest smoothing systematic uncertainties in the 5 added sites.**

The provided 3.4% number for median smoothing systematic uncertainty is the one given in Vigouroux et al., AMT, 2018. The 13% and 14% for the 5 added sites, are for the total systematic uncertainty (dominated by the spectroscopy), and not for the smoothing part only. To avoid the confusion, we have changed the sentence to:
*"For the five added sites, the median total systematic uncertainty is 13% (Jungfraujoch, Tsukuba, Palau), or 14% (Rikubetsu, Xianghe), commensurate with the other sites."*

**Page 8, line 25, please remove "so"**

Done.

**Page 11, line 29, would be nice if you include one or two sentences describing the main differences between OFFL, RPRO and NRTI products. Are they different at all.**

OFFL, RPRO and NRTI share the same algorithm (for the versions used in the paper). Changes of version numbers refer to changes in other components of the operational processor. However, slight differences come from auxiliary data. A priori profiles used for NRTI are from TM5 forecast model, while they are from TM5 analysis for OFFL/RPRO (this makes almost no difference since HCHO is not assimilated).
For the reprocessing (RPRO), data have been processed using 7-days parallelization (in order to speed up the reprocessing). It means that the slant columns used for the background correction are always at least 7 days older, while for OFFL and NRT, the gap is only 1 day. It results in stripes slightly more pronounced in the RPRO product than in the other versions.

We do not give these details on the OFFL / RPRO / NRTI data because they have negligible impact on the satellite data and validation results. But we have added a Table (Table 1 in the new version) with the date of the different versions, and all details can be found in the Readme file given as a reference in the manuscript.

**Page 16, line 31, would you please clarify how the collocation plays a role in Maïdo? Fire emissions are included in the calculation of the a-priori profiles? Could fire emissions enhanced the HCHO amounts? What is the effect of changing the collocation radius in this station?**

Maïdo is usually a clean site. During the biomass burning period, some plumes (mainly coming from Madagascar for the short lifetime species HCHO) can cross over Reunion Island. Since the overestimation of TROPOMI at Maïdo is larger during the biomass burning months, we suggest that this could be due to plumes that would be present in the 20km circle around the station covered by TROPOMI but not in the line of sight of the FTIR measurements for the collocated days. Unfortunately the collocation effect could not be confirmed at Maïdo: the 10km radius criterion lead to very few coincidences at Maïdo (see Fig. 1 of our Reply to referee#1), and none are during the biomass burning season.
This was only a suggestion from our side (we wrote "the collocation of the plumes *might* play a role there"), and could be investigated when more data are available in a future work.
